# Priming mobilization of hair follicle stem cells triggers permanent loss of regeneration after alkylating chemotherapy

Jin Yong Kim [1,2,3], Jungyoon Ohn [1,2,3], Ji-Seon Yoon[2,3], Bo Mi Kang[1,2,3], Minji Park[2,3,4], Sookyung Kim[4,5], Woochan Lee [4,5], Sungjoo Hwang[6], Jong-Il Kim [4,5], Kyu Han Kim [1,2,3] & Ohsang Kwon[1,2,3]

The maintenance of genetic integrity is critical for stem cells to ensure homeostasis and regeneration. Little is known about how adult stem cells respond to irreversible DNA damage, resulting in loss of regeneration in humans. Here, we establish a permanent regeneration loss model using cycling human hair follicles treated with alkylating agents: busulfan followed by cyclophosphamide. We uncover the underlying mechanisms by which hair follicle stem cells (HFSCs) lose their pool. In contrast to immediate destructive changes in rapidly proliferating hair matrix cells, quiescent HFSCs show unexpected massive proliferation after busulfan and then undergo large-scale apoptosis following cyclophosphamide. HFSC proliferation is activated through PI3K/Akt pathway, and depletion is driven by p53/p38-induced cell death. RNA-seq analysis shows that HFSCs experience mitotic catastrophe with G2/M checkpoint activation. Our findings indicate that priming mobilization causes stem cells to lose their resistance to DNA damage, resulting in permanent loss of regeneration after alkylating chemotherapy.

---

[1] Department of Dermatology, Seoul National University College of Medicine, Seoul 03080, Korea. [2] Institute of Human-Environment Interface Biology, Medical Research Center, Seoul National University, Seoul 03080, Korea. [3] Laboratory of Cutaneous Aging and Hair Research, Biomedical Research Institute, Seoul National University Hospital, Seoul 03080, Korea. [4] Department of Biomedical Sciences, Seoul National University College of Medicine, Seoul 03080, Korea. [5] Department of Biochemistry and Molecular Biology, Seoul National University College of Medicine, Seoul 03080, Korea. [6] Dr. Hwang's Hair-Hair Clinic, Seoul 06035, Korea. Correspondence and requests for materials should be addressed to O.K. (email: oskwon@snu.ac.kr)

In recent decades, alkylating agents have been widely used as chemotherapeutic treatments for cancer patients and have improved prognosis dramatically[1]. Chemotherapeutic alkylating agents induce a range of cytotoxic and mutagenic adducts directly on DNA, triggering the death of rapidly proliferating cancer cells but also affecting healthy normal cells[2,3]. Most alkylating agents were discovered based on their efficacy against cancer cells, so off-target effects on normal tissues are inevitable. Therefore, DNA damage from alkylating chemotherapy increases the risk of genetic mutations in normal tissues that can induce de novo cancer without a proper DNA repair process[4,5].

Adult stem cells govern their homeostasis and regeneration abilities throughout the lifespan in their specialized niches. To ensure proper stem cell function, the maintenance of accurate genetic information is critical. After exposure to alkylating agents, adult stem cells are at a critical decision point that can result in DNA damage repair to preserve regeneration capacity or programmed cell death to avoid propagating mutations[2,6]. Imbalances between these responses can result in deleterious consequences at the tissue and organism levels[2]. Understanding the intrinsic cellular response to DNA damage is particularly relevant for both cancer prevention and cancer therapy. However, little is known about how human adult stem cells functionally respond to DNA damage within their natural niches or how they eventually lose their regenerative ability to prevent genetic mutations.

Hair follicles (HFs) are representative regenerative organs in humans, and they undergo lifelong cyclic growth, consisting of the anagen (active growth), catagen (regression), and telogen (relative rest) phases[7,8]. The upper structure above the bulge is the permanent segment regardless of the hair cycle phase, whereas the lower structure below the bulge is the regenerative segment that shrinks to a minimum structure of the hair germ during the telogen phase[9]. In the bulge, quiescent HF stem cells (HFSCs) are constantly preserved as the reservoir for the next hair cycle[9]. During the early period of anagen onset, HFSCs proliferate and self-renew to generate cells that form the future hair germ[9]. To regenerate the anagen HF, more active hair germ cells proliferate first to support the formation of the hair bulb, followed by the activation of quiescent HFSCs, which contribute to the outer root sheath (ORS)[9,10]. Based on this supply mechanism of follicular epithelial cells (EPI), the lower segment expands dramatically into a cylindrical HF structure. The transient segment of anagen HFs, the bulb, is densely crowded with rapidly proliferating hair matrix (Mx) cells, without long-lived quiescent stem cells. Because the majority of human scalp HFs are in the anagen phase, the highly proliferative nature of the bulb makes anagen HFs one of the most sensitive organs to genotoxic injury[11].

HFSCs are generally less sensitive to chemotherapy-induced cell death due to intrinsic mechanisms for increasing their resistance to DNA damage, such as a low proliferation rate, higher expression levels of aldehyde dehydrogenase and anti-apoptotic proteins, and transient stabilization of p53[4,12]. These HFSC characteristics have been consistently shown in the recently introduced cycling human HF model, which is derived from the transplantation of microdissected human HFs into immunodeficient mice[13]. After administration of cyclophosphamide (Cy) to mice, human anagen HFs show two damage response pathways, namely, the dystrophic anagen (low-dose Cy, 100 mg/kg) and dystrophic catagen (high-dose Cy, 150 mg/kg) pathways[13]. Eventually, damaged HFs undergo full recovery as terminal anagen HFs in both groups, which is the key feature of the transient chemotherapy-induced alopecia (CIA) that is observed in cancer patients[12,13]. The quiescent HFSCs maintain their stem cell pool in the bulge without loss of regeneration capacity[13,14],

whereas the major target of DNA damage-induced apoptosis is the rapidly proliferating Mx cells in the bulb, which are capable of complete regeneration in the normal hair cycle.

In contrast, total loss or incomplete hair regrowth 6 months after the cessation of chemotherapy is defined as permanent CIA in cancer patients. Permanent loss of hair regeneration has been increasingly reported following the use of busulfan (Bu), Cy, melphalan, etoposide, thiotepa, carboplatin, docetaxel, and paclitaxel[15,16]. The most noxious regimen, especially for childhood cancer survivors, is Bu followed by Cy (designated Bu/Cy) as the conditioning regimen for hematopoietic stem cell transplantation (HSCT) in hematological malignancies[15,17–19].

Here, using cycling human HFs, we establish a model for permanent loss of regeneration with sequential Bu/Cy treatment. We uncover the underlying mechanisms by which adult stem cells eventually lose their pool respond to sequential DNA damage in their native niche. The quiescent HFSCs show reactive proliferation after priming Bu treatment and became vulnerable to DNA damage-induced cell death caused by following Cy treatment. DNA double-strand breaks in HFSCs are persistent in the permanent loss condition in contrast to being resolved in the transient loss condition. Bu treatment situationally induces activation or apoptosis of HFSCs depending on the proliferation status, and DNA damage-induced cell death is pronounced in the DNA-replicating S phase. Mobilization and depletion of HFSCs are consequences of the stepwise phase conversion of PI3K/Akt pathway activation followed by p53/p38-induced cell death. Global gene expression analysis of laser-captured microdissected in vivo HFSCs shows that HFSCs experience mitotic catastrophe with activation of the G2/M checkpoint.

## Results

**Permanent loss of HF regeneration after Bu/Cy treatment**. To define the pathological consequences of alkylation-induced DNA damage in human HFs, microdissected human HFs were transplanted into severe combined immunodeficiency hairless outbred mice. After 22 weeks, transplanted human HFs were regenerated into terminal HFs showing anagen stage 6 morphology, characterized by spindle-like dermal papilla (DP) and Y-shaped melanin granules (Fig. 1a)[13]. Based on the conditioning schedule for HSCT[20], 4 doses of Bu (20 mg/kg/day) and 2 doses of low-dose Cy (100 mg/kg/day)[13] were administered (designated Bu/Cy) to the mice, and subsequent analysis was conducted over 6 months (Fig. 1b). Histological examination showed extensive destructive changes mainly in the bulb after Bu/Cy treatment. Consequently, the HF epithelium and DP structure were completely lost without hair regrowth on day 60, indicating permanent loss of HF regeneration (Fig. 1c). Dystrophic melanin clumping was observed around the DP (Fig. 1d), and alkaline phosphatase activity was lost in the DP (Fig. 1e). Considering that the permanent segment of the HF was lost completely, this response pathway was distinct from the previously reported dystrophic anagen or catagen pathways[13]. Up to 6 months after the cessation of treatment, there was no evidence of hair regrowth or cycling human HFs inside or outside of the back skin of the mice. We established a proper model for permanent loss of HF regeneration using cycling human HFs subjected to sequential Bu/Cy treatment that closely reflects permanent CIA in humans.

**Distinct spatiotemporal response is orchestrated in the bulb**. To dissect the spatiotemporal response of human HFs after Bu/Cy treatment, the bulb area was divided into three distinctive zones: (1) rapidly proliferating Mx cells, (2) relatively quiescent ORS cells, and (3) DP cells (Fig. 2). The number of p53[+] DNA-

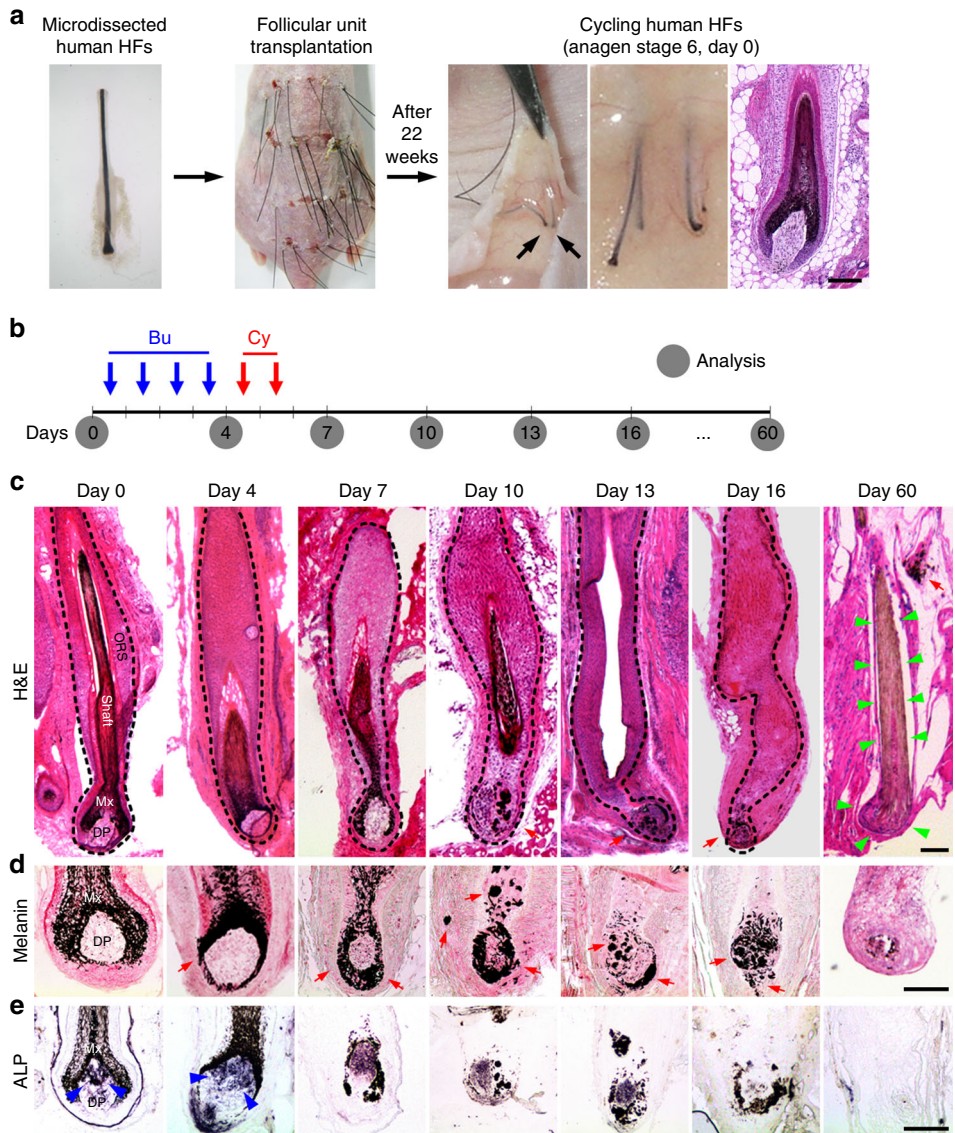

**Fig. 1** Permanent loss of HF regeneration after Bu/Cy treatment. **a** Experimental model of cycling human HFs for the investigation of permanent loss of regeneration after alkylating chemotherapy. Microdissected human HFs were transplanted into severe combined immunodeficiency hairless outbred mice (n = 765 transplanted HFs, 32 mice). After 22 weeks of transplantation, human HFs were regenerated into terminal HFs showing anagen stage 6 morphology. **b** Schedule for sequential Bu/Cy treatment (Bu 20 mg/kg/day, Cy 100 mg/kg/day) into mice with transplanted human HFs, followed by analysis of human HFs at 0, 4, 7, 10, 13, 16, and 60 days. **c** Representative images of destructive changes mainly in the bulb after Bu/Cy treatment (n = 397 transplanted HFs, 17 mice, 3 independent experiments). The HF epithelium and the DP structure were lost (green arrowhead) with no hair regrowth on day 60, indicating that the permanent segment of HF was lost completely (H&E). **d** Representative images of dystrophic melanin clumping (red arrow) around the DP in the bulb after Bu/Cy treatment. The pigment distribution became more disorganized over time (Masson Fontana). **e** Representative images of ALP activity (blue arrowhead) in the DP after Bu/Cy treatment. The ALP activity completely disappeared after day 7 (scale bar = 100 μm). Bu busulfan, Cy cyclophosphamide, H&E hematoxylin and eosin, HF hair follicle, Mx hair matrix, ORS outer root sheath, DP dermal papilla, ALP alkaline phosphatase

damaged cells was significantly increased in the EPI (combined zone of Mx and ORS) after Bu treatment (on day (4) and peaked after Bu/Cy treatment (on day 7; Fig. 2a). Fas+ and TUNEL+ apoptotic cells already appeared in the Mx on day 4, but an explosive increase was observed in the ORS on day 7 (Fig. 2b and c). There were many Ki67+ proliferating cells pushing the hair shaft out in the Mx, but these cells rapidly disappeared on day 4. Unexpectedly, in the basal ORS layer, a few newly proliferating cells emerged on day 4, and these cells were immediately quenched on day 7 (Fig. 2d). HF melanocytes located in the bulb showed a pattern similar to that of the Mx cells, as Tyr+ and TRP1+ cells started to fade away on day 4 and then disappeared on day 7 (Supplementary Fig. 1). Taken

together, these data show that the rapidly proliferating Mx cells underwent large-scale apoptosis immediately after Bu treatment, while the relatively quiescent basal ORS cells showed reactive proliferation after Bu treatment and were then mostly quenched by Bu/Cy treatment. Considering that Bu is one of the oldest alkylating agents and causes cytotoxicity through an alkylating reaction[21], Bu-induced proliferation of basal ORS cells was an noteworthy observation (Supplementary Fig. 2).

**Priming proliferation precedes loss of stem cell in the bulge.** To clarify the consequences of chemotherapy in HFSCs, we revisited the previous transient loss model[13] compared to this permanent loss model (Fig. 3a). For the transient loss condition, a single dose

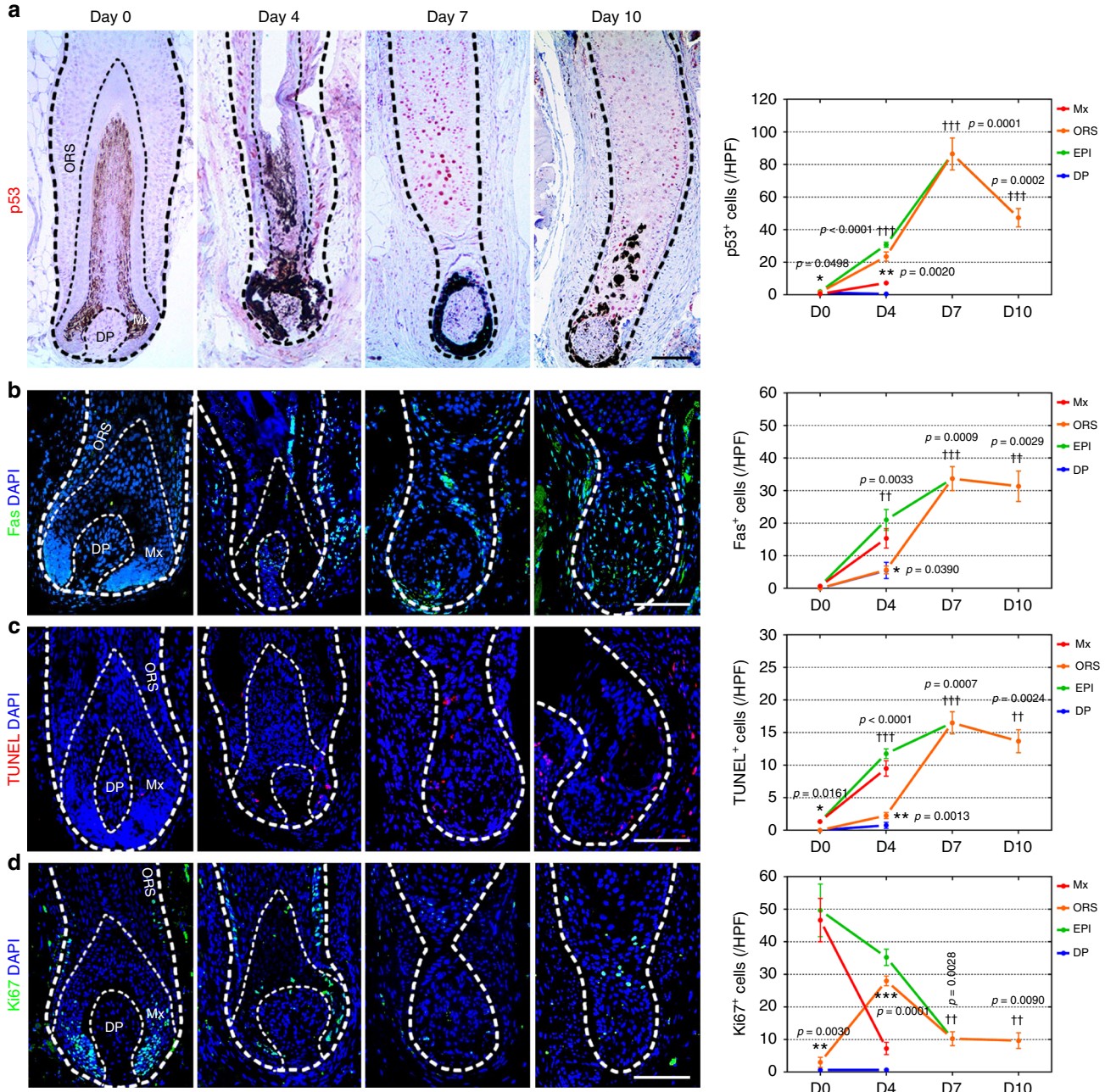

**Fig. 2** Distinct spatiotemporal response after Bu/Cy treatment in the bulb. **a** Representative images and quantification of p53 staining to detect DNA-damaged cells in the bulb after Bu/Cy treatment ($n = 4, 4, 4$, and 3 biological replicates/timepoint; immunoperoxidase). The number of p53[+] DNA-damaged cells was increased in the Mx and ORS as time progressed (day 0 to 7). Representative images and quantification of **b**, Fas ($n = 3, 3, 3$, and 3 biological replicates/timepoint) and **c**, TUNEL ($n = 3, 4, 4$, and 3 biological replicates/timepoint) staining to detect apoptotic cell death in the bulb after Bu/Cy treatment. The number of apoptotic cells increased earlier in the Mx (day 0 to 4) than in the ORS (day 4 to 7). **d** Representative images and quantification of Ki67 staining to detect cellular proliferation in the bulb after Bu/Cy treatment ($n = 3, 4, 4$, and 3 biological replicates/timepoint). The number of Ki67[+] cells decreased in the Mx and EPI. Note that the basal ORS cells showed remarkable proliferation after Bu treatment (day 0 to 4) (immunofluorescence; scale bar = 100 μm). Bu busulfan, Cy cyclophosphamide, Bu/Cy busulfan followed by cyclophosphamide, HF hair follicle, HPF high-power field, Mx hair matrix, ORS outer root sheath, EPI epithelium (combined zone of Mx and ORS), DP dermal papilla. Data are mean ± SEM. Source data are provided as a Source Data file. *$p < 0.05$ (ORS vs. Mx); **$p < 0.01$ (ORS vs. Mx); ***$p < 0.001$ (ORS vs. Mx, unpaired $t$ test); [†]$p < 0.05$ (EPI vs. D0); [††]$p < 0.01$ (EPI vs. D0); [†††]$p < 0.001$ (EPI vs. D0, unpaired $t$ test)

of Cy (150 mg/kg/day) was administered (designated Cy only) to mice with cycling human HFs. In the bulge of control HFs, few Ki67[+] proliferating cells are located in the K15[−] suprabasal layer, while HFSCs remain quiescent in the K15[+] basal layer. Remarkably, HFSCs showed large-scale proliferation after Bu treatment, and this proliferation was completely quenched after

Bu/Cy treatment (Fig. 3b). In the transient loss condition, p53[+] cells were observed after Cy only treatment in the suprabasal layer, which had been a proliferative zone in control HFs[13]. However, in the permanent loss condition, lining p53[+] cells emerged after Bu/Cy treatment in the basal layer, which had been a proliferative zone when after Bu treatment (Fig. 3c).

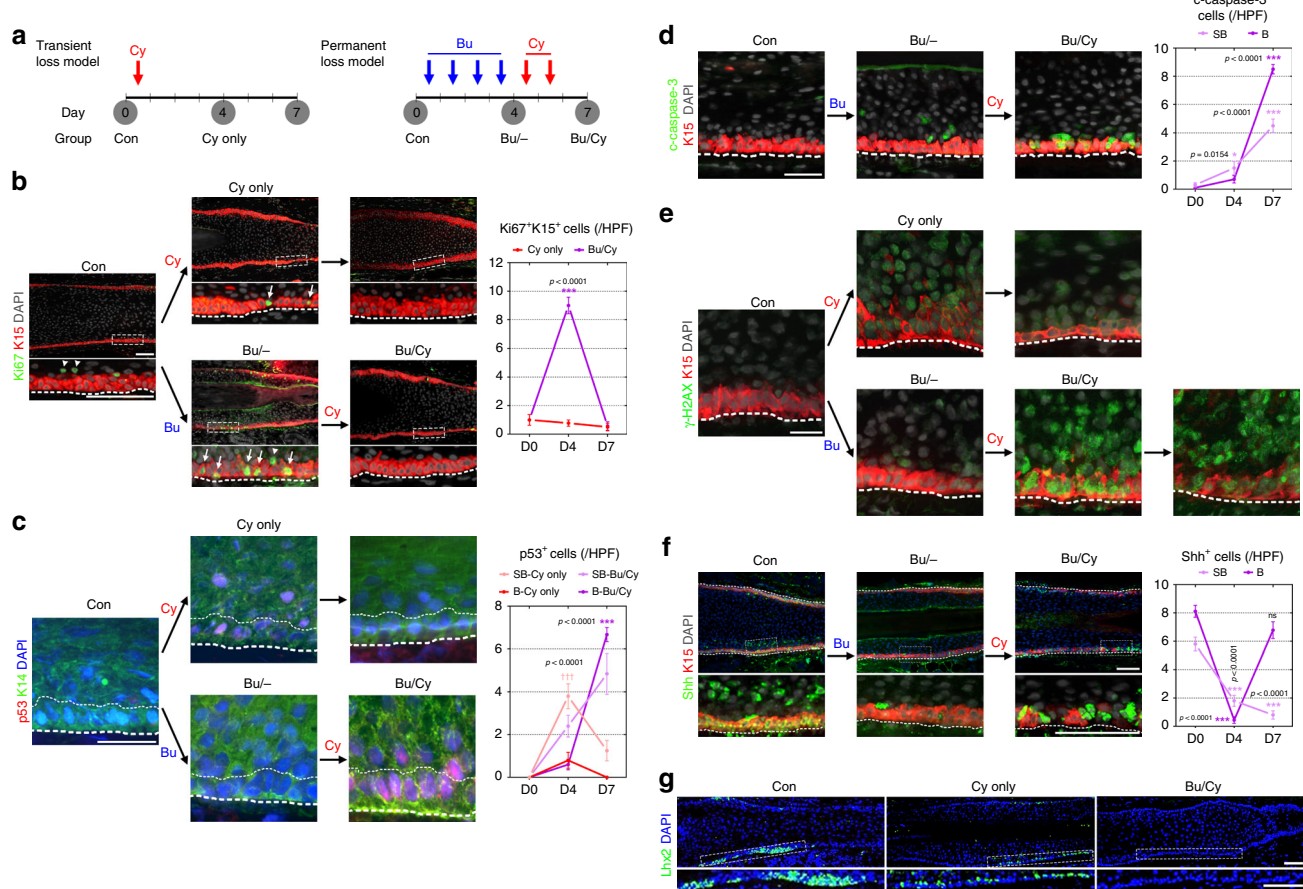

**Fig. 3** Priming proliferation precedes loss of stem cell reserve in the bulge. **a** Experimental models for transient loss after Cy only treatment vs. permanent loss after Bu/Cy treatment. **b** Representative images and quantification of Ki67+ cells among K15+ HFSCs in the bulge ($n = 8$ (D0 Con), 9 (D4 Cy only), 8 (D7 Cy only), 6 (D4 Bu), and 7 (D7 BuCy) biological replicates). Few Ki67+ cells were observed in the K15− suprabasal layer in control HFs (white arrowhead). HFSCs showed large-scale proliferation after Bu treatment (white arrow), and this proliferation was completely quenched after Bu/Cy treatment. **c** Representative images and quantification of p53+ cells in the K14+ HF epithelium in the bulge ($n = 6$ (D0 Con), 5 (D4 Cy only), 4 (D7 Cy only), 5 (D4 Bu), and 6 (D7 BuCy) biological replicates). p53+ cells were observed in the suprabasal layer after Cy only treatment, but lining p53+ cells emerged in the basal layer after Bu/Cy treatment. **d** Representative images and quantification of c-caspase-3+ cells among K15+ HFSCs in the bulge ($n = 10$ biological replicates/timepoint). Several c-caspase-3+ cells were detected in the K15+ basal layer after Bu/Cy treatment. **e** Representative images for γ-H2AX+ cells among K15+ HFSCs in the bulge. γ-H2AX+ cells were massively detected and persistent in the basal and suprabasal layers after Bu/Cy treatment. **f** Representative images and quantification of Shh+ cells among K15+ HFSCs in the bulge ($n = 9$ biological replicates/timepoint). Shh+ cells disappeared after Bu treatment and then reappeared without K15 expression after Bu/Cy treatment in the basal layer. **g** Representative images of Lhx2+ cells in the bulge on day 13 after alkylating chemotherapy (immunofluorescence; scale bar = 100 μm in **b**, **f**, **g**; 50 μm in **c**, **d**, **e**). Bu busulfan, Cy cyclophosphamide, Bu/Cy busulfan followed by cyclophosphamide, SB suprabasal layer, B basal layer, HF hair follicle, HPF high-power field, HFSC hair follicle stem cell, c-caspase-3 cleaved caspase-3. Data are mean ± SEM. Source data are provided as a Source Data file. ***$p < 0.001$ (Cy only vs. Bu/Cy), †††$p < 0.001$ (SB vs. B, two-way analysis of variance with Sidak's multiple comparisons test) in **b** and **c**; *$p < 0.05$ (vs. D0), ***$p < 0.001$ (vs. D0, two-way analysis of variance with Sidak's multiple comparisons test) in **d** and **f**

Consequently, HFSCs underwent large-scale apoptosis through the activation of caspase-3 in the K15+ basal layer, showing spatiotemporal transitions from the proliferative zone into the apoptotic zone in the bulge area (Fig. 3d).

Then, γ-H2AX expression was assessed to identify actual DNA damage, which serves as a marker of double-strand breaks. H2AX phosphorylation occurred mostly in the suprabasal layer after Cy only treatment and then seemed to resolve over time. However, γ-H2AX+ cells were massively detected and persistent in the basal and suprabasal layers after Bu/Cy treatment, indicating unrepaired DNA damage (Fig. 3e). Sonic hedgehog (Shh) signaling blocks differentiation into progeny in quiescent HFSCs in humans[22] and is suggested to have a role in chemotherapy-induced alopecia[23]. In control HFs, Shh expression was observed in HFSCs in the basal layer and in the second

progeny in the suprabasal layer, which are more differentiated cells than the first progeny of HFSCs. Intriguingly, Shh+ cells almost disappeared after Bu treatment and then reappeared without K15 expression in the basal layer after Bu/Cy treatment (Fig. 3f). To assess the stem cell reserve, Lhx2 expression was assessed on day 13 after chemotherapy, which is the master transcription factor for HFSC stemness[24]. There were no detectable Lhx2+ cells in the basal bulge in Bu/Cy-treated HF remnants (Fig. 3g)[24]. Taken together, these results suggest that HFSCs that lost their quiescence after Bu treatment showed a differentiated phenotype in addition to massive apoptosis after Bu/Cy treatment, indicating loss of the stem cell reserve. In conclusion, human HFSCs are resistant to Cy treatment when they are quiescent under normal circumstances[13,25]. However, after mobilizing signals from priming Bu treatment, HFSCs

become vulnerable to DNA damage from subsequent Cy treatment, leading to apoptotic depletion and loss of regenerative potential.

**Bu-induced proliferation followed by Cy-induced apoptosis.** To dissect the molecular cascade after Bu/Cy treatment, a controllable ex vivo and in vitro model was established using active molecules of alkylating agents. In the absence of a hepatic cytochrome P450 system, 4-hydroperoxycyclophosphamide (4-HCy) is an appropriate active metabolite for Cy, generating the downstream active toxic component in aqueous solution[26]. Despite the different clinical dosages of the two drugs (Bu 4 mg/kg/day; Cy 60 mg/kg/day), the plasma concentrations of the active molecules are pharmacokinetically within a similar range (Bu, 600–900 ng/mL; 4-OHCy, 436–1140 ng/mL) in humans[20,27]. The minimal threshold concentrations (Bu, 50 µmol/L; 4-HCy, 40 µmol/L) were tested in ex vivo organ-cultured human HFs and adopted in further experiments (Fig. 4a). Hair shaft elongation, which reflects the activity of Mx cells in the bulb, slowed after Bu treatment and halted after Bu/Cy treatment (Fig. 4b). In hair cycle score analysis[28], Bu-treated HFs prematurely entered catagen stage without p53-dependent apoptosis in their epithelial strands (Supplementary Fig. 3), suggesting that HF epithelial cells are still biologically viable enough to progress into their response pathway. However, Bu/Cy-treated HFs showed a completely shrunken bulb morphology, indicating a complete loss of the bulb and cellular arrest of the HF epithelium (Fig. 4c and d). Consistent with the in vivo HF xenografts, Ki67+ proliferating cells appeared remarkably in the K15+ basal bulge after Bu treatment and then immediately disappeared after Bu/Cy treatment (Fig. 4e). This change represented a large amount of p53-dependent apoptosis in the basal bulge (Fig. 4f), as indicated by the coexpression of p53 and cleaved caspase-3 (Fig. 4g). This ex vivo reproduction of the phase conversion of DNA damage response after Bu/Cy treatment allowed us to determine the molecular mechanism of permanent loss of HFSCs.

**DNA damage responses depending on proliferation status.** To assess this cell cycle-dependent vulnerability to genotoxicity, we analyzed the cellular responses of human ORS cells according to the proliferation status (Fig. 5a). To closely simulate HFSCs in vitro, holoclone-rich ORS cells were directly derived from the bulge of human HFs and divided into two different statuses: actively growing and confluent quiescent at early stages[29]. The quiescent status was induced by allowing the cells to reach 100% confluence, not by serum deprivation, for the appropriate conditions allowing cells recover from DNA damage[30]. By flow cytometry for ORS cell markers (CD29, CD49f, CD133, and CD200), actively growing cells (39% in S phase) and confluent quiescent cells (9% in S phase) were analyzed as homogenous populations, except for their S phase cell percentages (Supplementary Fig. 4). Bu treatment reduced the S phase subset in growing cells but induced a remarkable increase in the S phase subset in quiescent cells. Interestingly, Cy treatment resulted in an increase in the S phase subset in quiescent cells, which is suggested to represent S phase arrest (Fig. 5b). Next, the outcome of sequential Bu/Cy treatment was assessed in quiescent ORS cells. Based on the time span of the human cell cycle[31], cells were treated with Cy when they were maximally in the S phase after Bu priming (Fig. 5c). The final number of viable ORS cells markedly increased in the Bu only-treated group but almost disappeared in the Bu/Cy-treated group. Concordantly, a significant amount of cell debris was detected in the Bu/Cy-treated group, indicating massive cell death (Fig. 5d). Thus, the S phase-

dependent change in quiescent ORS cells demonstrated reactive proliferation after Bu treatment and subsequent cell death caused by Bu/Cy treatment. This result also suggests that human HFSCs are more sensitive to alkylation-induced DNA damage during their proliferative status.

**PI3K/Akt pathway activation and p53/p38-induced cell death.** To identify the molecular mechanism of HFSC proliferation and apoptosis, we analyzed the effects of alkylating chemotherapy on the PI3K/Akt pathway and the p53/p38-induced cascade. Holoclone-rich ORS cells were harvested after 1, 3, 6, and 12 h of Bu and/or Cy treatment (Fig. 6a). The Bu only-treated cells showed no significant changes in their morphology. However, the Cy-treated cells showed gradual changes, including membrane undulation, cellular shrinkage, and ultimately, detachment from and floating in the dish (Supplementary Fig. 5). Protein analysis of Bu-treated cells showed a remarkable increase in PI3K and Akt phosphorylation, with upregulated cyclin D1 promoting the G1/S phase transition through the PI3K/Akt pathway[32] (Fig. 6b, Supplementary Figs. 6 and 7). Protein analysis of Cy-treated cells showed a sustained increase in p53 and p38 phosphorylation, which is critical for chemotherapy-induced apoptosis and mitotic catastrophe[33], with downregulated p21 in the BuCy-treated cells, which is an on/off switch for cell survival vs. cell death in the p53 decision-making process[34–36]. To determine the temporal dynamics after sequential Bu/Cy treatment, ORS cells were consecutively harvested after 0, 1, 3, and 6 h of priming Bu treatment and then after 1, 3, 6, and 12 h of subsequent Cy treatment (Fig. 6c). Protein analysis showed dramatic temporal changes in two different phases: the PI3K/Akt pathway activation with upregulated cyclin D1 after Bu treatment, and the subsequent conversion into p53/p38-induced cell death with PI3K/Akt pathway inhibition after Bu/Cy treatment, resulting in dynamic changes in p21, representing the cellular decision of cell survival or cell death (Fig. 6d).

To confirm the phase conversion of the DNA damage response, we performed immunostaining for the phosphorylated Akt and p38 proteins in the in vivo HF xenografts. Consistent with the protein analysis, Akt phosphorylation occurred after Bu treatment (Fig. 7a), and p38 phosphorylation occurred after Bu/Cy treatment (Fig. 7b) in the K15+ HFSCs.

Then, we investigated the roles of the PI3K/Akt pathway and the p38-induced cascade with specific inhibitors of PI3K (20 µM; LY294002) and p38 (10–20 µM; SB202190 or SB203580). Under treatment with the PI3K inhibitor, Akt phosphorylation was blocked, with loss of cyclin D1 induction (Fig. 7c and Supplementary Fig. 8a). Notably, p53 and phosphorylated p38 were upregulated during blockade of the PI3K/Akt pathway, supporting that the PI3K/Akt pathway is the active part of DNA repair process[36]. Unexpectedly, the driving force of p38 phosphorylation after alkylating chemotherapy was too strong to be blocked by treatment with p38 inhibitors (Fig. 7d and Supplementary Fig. 8b)[37]. Overwhelming potentiation of p38 phosphorylation was observed, with a DNA damage severity-dependent downregulation of p21 and a marginal increase in cleaved caspase-3. The roles of the PI3K/Akt pathway and p53/p38-induced cascade were clearly observed during phase conversion after sequential Bu/Cu treatment (Fig. 7e, Supplementary Figs. 9 and 10). The temporal changes in p53 combined with phosphorylated Akt, cyclin D1, phosphorylated p38, p21, and cleaved PARP directly demonstrated two points; the first is that DNA damage sensitivity remains low if there is no priming proliferation by a PI3K inhibitor, and the other is that cells with DNA damage are not efficiently eliminated by insufficient p38 activation even at the last timepoint.

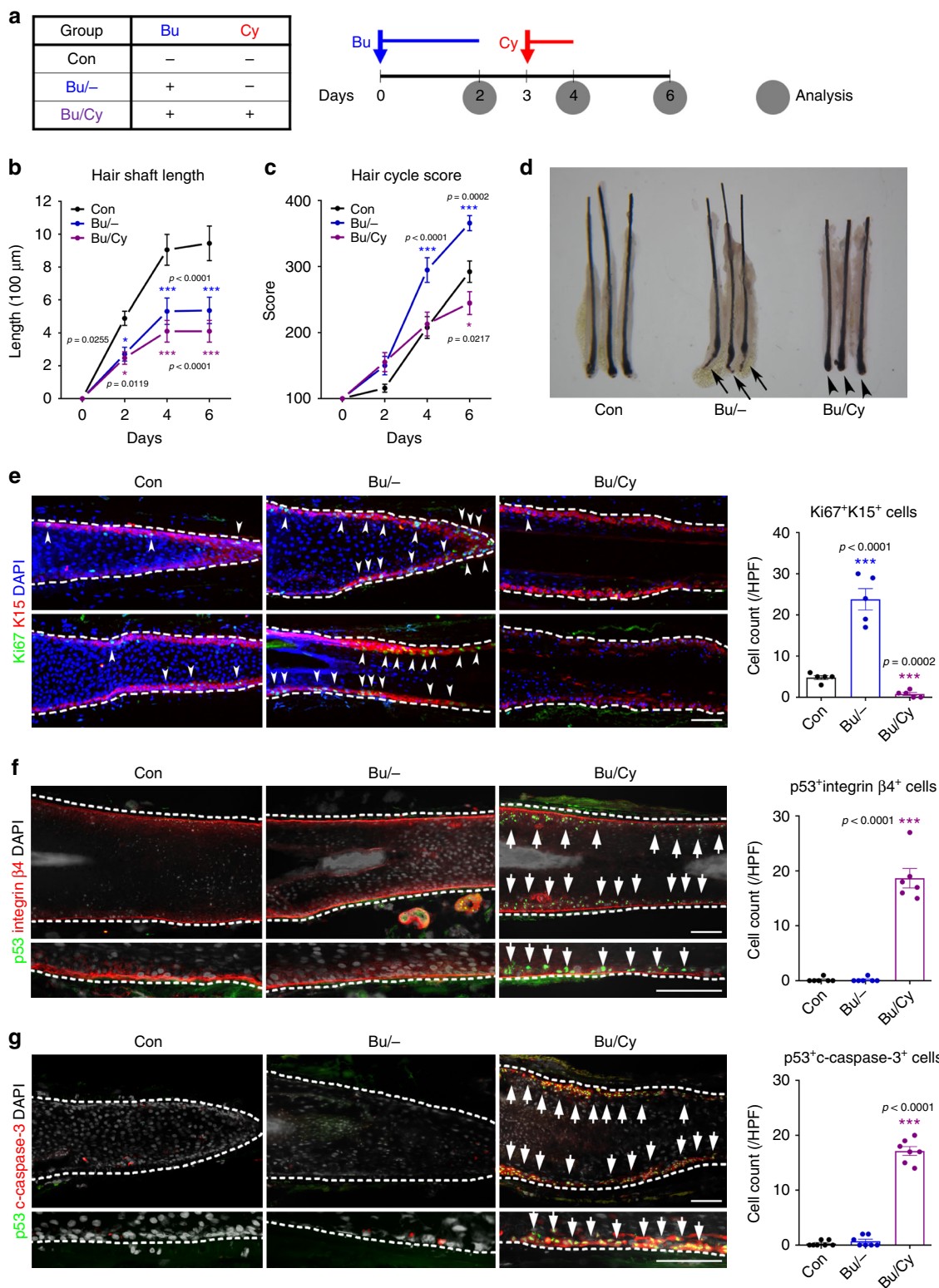

**Global gene expression changes of in vivo HFSCs**. To consolidate the HFSC response in permanent CIA in humans, global gene expression changes were evaluated in in vivo HF xenografts after sequential Bu/Cy treatment. Using laser capture microdissection, HFSCs were obtained from the basal layer in the bulge and analyzed with RNA-seq (Supplementary Fig. 11)[38,39]. Based on hierarchical clustering (Fig. 8a), the upregulated genes after Bu or Bu/Cy treatment were selected for functional enrichment pathway analysis from the Reactome database[40]. Mitotic cell cycle

pathways and their relevant genes were substantially upregulated after Bu treatment, accompanied by the DNA repair pathway and p53 transcriptional regulation (Fig. 8b and Supplementary Fig. 12a). Intriguingly, cell cycle checkpoint pathways, especially the G2/M checkpoint and related genes were upregulated after Bu/Cy treatment, along with mitochondrial membrane-associated pathways (Fig. 8c and Supplementary Fig. 12b). This result indicates that the majority of HFSCs experienced mitotic catastrophe and stalled in S phase with activation of the G2/M

**Fig. 4** Bu-induced proliferation followed by Cy-induced apoptosis in organ-cultured HFs. **a** Schedule for Bu/– and Bu/Cy treatments and subsequent analysis at 2, 4, and 6 days after alkylating chemotherapy. **b** Hair shaft elongation and **c**, hair cycle score of organ-cultured human HFs after alkylating chemotherapy ($n = 38$ biological replicates/timepoint, 3 independent experiments). Note that the score of the Bu/Cy group was lower than that of the control group on day 6. **d** Representative images of organ-cultured HFs of experimental groups. Control HFs maintained anagen morphology, and Bu-treated HFs prematurely entered catagen stage (black arrow); however, Bu/Cy-treated HFs showed completely shrunken bulb morphology (black arrowhead). Representative images and quantification of **e**, Ki67+ cells among K15+ HFSCs ($n = 5$ biological replicates/group), **f** p53+ cells among integrin β4+ basal cells ($n = 6$ biological replicates/group), and **g**, p53+ c-caspase-3+ cells ($n = 7$ biological replicates/group) in the bulge after alkylating chemotherapy. HFSCs showed remarkable cellular proliferation after Bu treatment (white arrowhead) and were completely quenched with large-scale apoptosis after Bu/Cy treatment (white arrow; immunofluorescence; scale bar = 100 μm). Bu busulfan, Cy cyclophosphamide, Bu/Cy busulfan followed by cyclophosphamide, HF hair follicle, HPF high-power field, HFSC hair follicle stem cell, c-caspase-3 cleaved caspase-3. Data are mean ± SEM. Source data are provided as a Source Data file. *$p < 0.05$ (vs. Con); ***$p < 0.001$ (vs. Con, two-way analysis of variance with Dunnett's multiple comparisons test) in **b** and **c**; ***$p < 0.001$ (vs. Con, unpaired $t$ test) in **e**, **f**, and **g**

checkpoint, which is consistent with the S phase arrest observed during the ORS cell response depending on proliferating status[41,42]. Gene expression of Ki67 (*MKI67*) and Survivin (*BIRC5*), an apoptosis inhibitor[42], was upregulated after Bu treatment and subsequently decreased after Bu/Cy treatment (Fig. 8d). Genes for maintaining HFSC stemness (*LHX2*, *PHLDA1*, *FZD1*, and *TGFB2*) were already downregulated after Bu treatment, supporting the role of priming proliferation in loss of the stem cell reserve[38,43].

## Discussion

Hair loss after chemotherapy is a serious problem for cancer survivors, causing not only psychosocial stress but also chemotherapy refusal due to fear of hair loss[17,44]. Undoubtedly, permanent CIA is a terrible aftereffect, impairing social activities during the remaining lifetime, especially for childhood cancer survivors[17]. Nevertheless, whether permanent CIA diagnosed in cancer survivors is in fact permanent or irreversible has yet to be determined because the mechanism by which the chemotherapy-resistant HFSC pool is depleted is not understood[17,45,46]. The absence of pathophysiological mechanisms has made it difficult to develop proper preventive or therapeutic options. Considering that an overdose of chemotherapy causes permanent loss of the host, which is more than just a failure of HF regeneration, it is essential to prepare optimal experimental models that reflect human HFSC biology with a physiologically relevant chemotherapy dosage. In this study, we establish in vivo, ex vivo, and in vitro models of the permanent loss of HF regeneration and provide multilevel biological evidence to unravel the underlying mechanism by which the HFSC pool can be depleted after alkylating chemotherapy.

To ensure tissue homeostasis or regeneration, DNA damage in adult stem cells is rapidly sensed and repaired by diverse mechanisms[4]. Depending on the extent of DNA damage, the rapidity of DNA repair, the stage of the cell cycle, and the strength of p53 activation, DNA-damaged cells can either undergo DNA damage repair to promote cell survival or programmed cell death to avoid propagating inaccurate genetic information[4]. Alkylating agents react with the electron-rich nitrogen or oxygen atoms of DNA bases to generate a variety of covalent adducts[47]. Cy is a bifunctional SN1-type alkylating agent with two active bischloroethyl groups that can react with separate bases in DNA to form interstrand crosslinks in addition to monoadducts or intrastrand crosslinks. Bu is a unique SN2-type alkylating agent with alkyl sulfonate groups that can induce monoadducts or intrastrand crosslinks[48]. Compared to the wide application range of Cy[47], Bu shows selective toxicity for early myeloid precursors, which is responsible for an important component of bone marrow ablative regimens[49,50]. The repair mechanisms for an adduct on a single DNA base include base excision repair and nucleotide excision repair[2,4]. However, for the

repair of interstrand crosslinks, it is necessary to coordinate complicated elements of the Fanconi anemia pathway, homologous recombination and translesion DNA synthesis[51]. Therefore, interstrand crosslinks are highly toxic DNA lesions that prevent replication and transcription by inhibiting DNA strand separation[51]. The interstrand crosslinks lead to cell death by p53-dependent and Fas ligand-dependent apoptosis or p53-independent mitotic catastrophe[51], which is a type of cell death that occurs during mitosis as a result of extensive DNA damage coupled with cell-cycle checkpoints and mitochondrial membrane permeabilization[42]. In this study, Bu/Cy-treated HFSCs not only undergo large-scale apoptosis through the p53/p38-induced cascade in the absence of p21 induction but also experience massive mitotic catastrophe with activation of the G2/M checkpoint showing S phase arrest.

The general assumption is that sensitivity to DNA damage-induced cell death is related to proliferation status[52]. In the hematopoietic system, quiescent hematopoietic stem cells (HSCs) generate all blood cell lineages via a highly proliferative amplifying progenitor that corresponds to epithelial cell lineages derived from HFSCs[53]. Human umbilical cord blood HSCs, which are highly proliferative, show induction of apoptosis and overt stem cell depletion with a slower rate of DNA repair after irradiation[54]. In contrast, mouse adult HSCs, which are mostly quiescent, show overt cell survival by allowing p53-mediated p21 induction to induce transient growth arrest and permit DNA repair[55]. Correspondingly, mouse adult HFSCs have been shown to be resistant to irradiation-induced cell death while quiescent in their native niche[25,34,56]. In this study, human HFSCs show opposing responses to DNA damage depending on the proliferation status. After Bu treatment, human HFSCs undergo reactive proliferation when they are quiescent in confluence or in their niche in vivo but undergo p53-induced apoptosis when they are actively proliferating or when they have committed to proliferative progeny in vivo. Then, proliferating HFSCs lose their resistance to DNA damage and prone to the apoptotic cascade after Cy treatment, indicating that priming proliferation makes HFSCs vulnerable to genotoxicity (Fig. 9).

The reactive proliferation of HFSCs can be understood as cellular attempts at DNA repair and tissue regeneration. The PI3K/Akt pathway, responsible for the mobilization of HFSCs[57], is a direct participant coregulated tightly with other components of DNA damage repair pathways[36,58]. For tissue regenerative attempts, mouse adult HFSCs exhibit slower activation dynamics only when HFs are severely damaged after ionizing radiation[59]. The priming activation of HFSCs is reminiscent of that observed in ovarian primordial follicles after a single dose of alkylating agent[60]. Dormant primordial follicles representing the "ovarian reserve" of potential fertility are recruited to undergo premature growth and differentiation, resulting in the depletion of stem cell reservoirs by activation of the PI3K/Akt pathway[60]. Considering

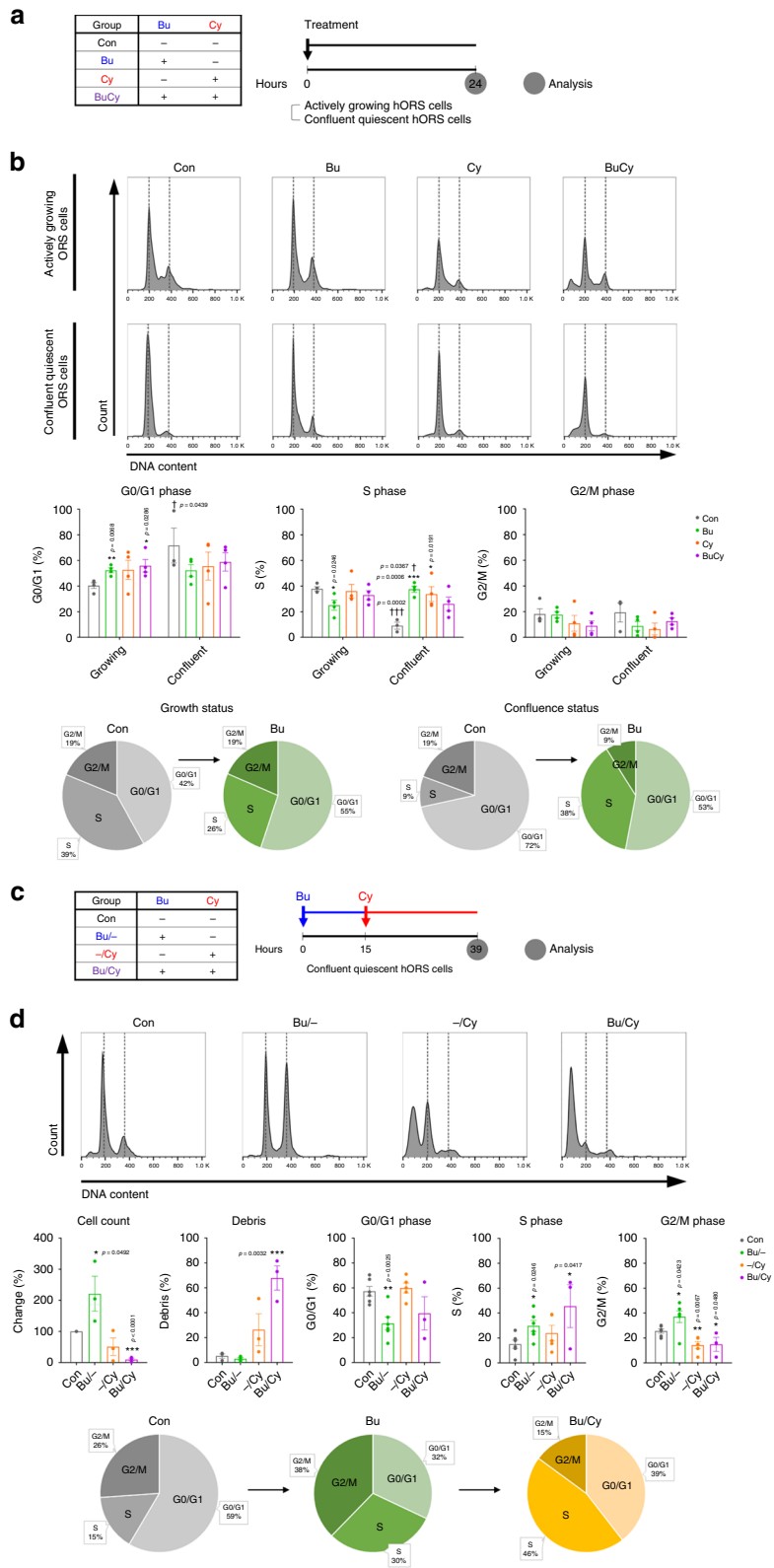

aging-dependent HFSC depletion is also triggered by DNA damage-induced loss of stemness[61], Bu-induced HFSC activation itself is thought to be a significant contributor to loss of stem cell reserve.

p53-dependent apoptosis is widely believed to be an essential player for the preservation of genomic integrity in response to DNA damage in the HF epithelium[56]. In addition, p38 kinase

activation is induced by DNA cross-linking agents and is sustained for more than a few days, triggering the apoptotic cascade[33,62–64]. p38-induced apoptosis can be driven not only by p53-dependent manner[63] but also in a p53-compromised state which originates from the overwhelming extent of DNA damage[64]. More interestingly, initiation of the G2/M checkpoint requires the activation of p38 kinase, and this p38-mediated

**Fig. 5** Cellular response to alkylating agents depending on proliferation status. **a** Schedule for single or combined treatment with alkylating agents and subsequent cell cycle analysis of actively growing and confluent quiescent ORS cells after 24 h of treatment. **b** Representative flow cytometry plots with propidium iodide staining and quantification of cell cycle analysis measured as the percentage of the total cell population of actively growing or confluent quiescent ORS cells ($n = 4$ biological replicates). Note that the Bu treatment reduced the S phase subset in growing cells but induced the S phase subset in quiescent cells. **c** Schedule for sequential treatment with alkylating agents and subsequent cell cycle analysis of confluent quiescent ORS cells after 15 plus 24 h of treatment. **d** Representative flow cytometry plots with propidium iodide staining and quantification of cell cycle analysis measured as the percentage of the total cell population of confluent quiescent ORS cells ($n = 6$ biological replicates), cell count and cell debris ($n = 3$ biological replicates). The initial Bu treatment forced the transition of quiescent cells into the S phase, and the subsequent Cy treatment eradicated these DNA-replicating cells, resulting in reduced total cell counts and increased cell debris. Bu busulfan, Cy cyclophosphamide, BuCy busulfan combined with cyclophosphamide, Bu/Cy busulfan followed by cyclophosphamide. Data are mean ± SEM. Source data are provided as a Source Data file. *$p < 0.05$ (vs. Con); **$p < 0.01$ (vs. Con); ***$p < 0.001$ (vs. Con, unpaired $t$ test); †$p < 0.05$ (Confluent vs. Growing); †††$p < 0.001$ (Confluent vs. Growing, unpaired $t$ test)

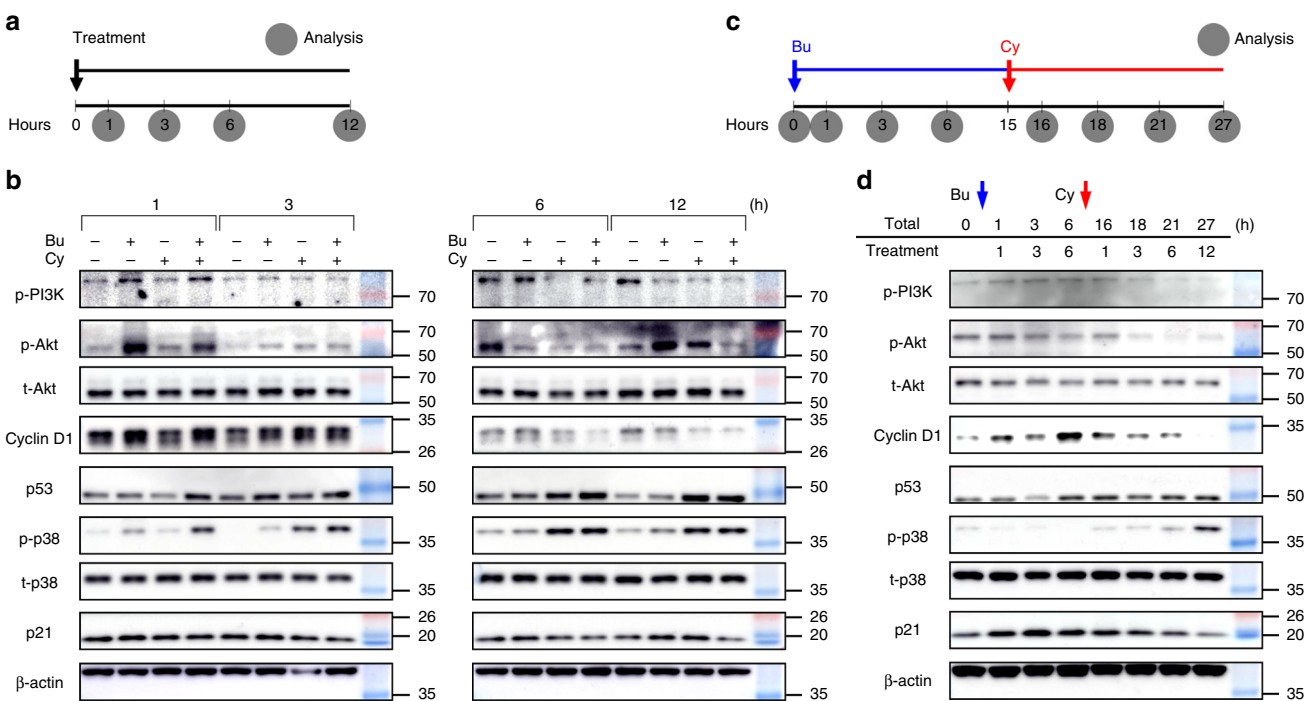

**Fig. 6** Bu and Cy trigger key signaling pathways for proliferation and cell death. **a** Schedule for Bu and/or Cy treatment and subsequent protein analysis after 1, 3, 6, and 12 h of treatment. **b** Protein analysis after Bu and/or Cy treatment comparing the concentrations of phosphorylated PI3K, phosphorylated and total Akt, cyclin D1, p53, phosphorylated and total p38, and p21, with β-actin as a loading control ($n = 6$ biological samples, Supplementary Fig. 7). **c** Schedule for sequential Bu/Cy treatment and subsequent protein analysis after 0, 1, 3, 6, 16, 18, 21, and 27 h of treatment. **d** Protein analysis after sequential Bu/Cy treatment comparing the concentrations of phosphorylated PI3K, phosphorylated and total Akt, cyclin D1, p53, phosphorylated and total p38, and p21, with β-actin as a loading control ($n = 4$ biological samples; Western blotting). Bu busulfan, Cy cyclophosphamide, Bu/Cy busulfan followed by cyclophosphamide

G2/M checkpoint also occurs in p53-deficient cells[65]. In this study, it is supposed that p53 activation is partially responsible for the apoptotic loss of HFSCs during the early period after DNA damage, but sustained p38 activation augmented the resulting cell death as a major mediator of mitotic catastrophe after irreversible DNA damage.

The Bu/Cy conditioning regimen shows superior anti-leukemic and immunosuppressive efficacy but exhibits severe off-target toxicity in host tissues. An animal study that examined the efficacy of bone marrow ablation according to the administration order of Bu and Cy, namely, "conventional" Bu/Cy and "modified" Cy/Bu, provided an interesting insight regarding the mechanism underlying the permanent loss of adult stem cells[66]. On the day of HSCT, the bone marrow cellularity decreases by 94% in Bu/Cy-treated mice, whereas this reduction is approximately 61% in mice receiving Cy/Bu, which means conditioning is not sufficient[66]. The Bu/Cy regimen is apparently more effective in eradicating host HSCs, so the administration order of Bu

and Cy has scientific significance in terms of the depletion of adult stem cells in vivo[66]. The insight that we can obtain in this study is that the forced mobilization of the quiescent stem cell population by a priming cue is suggested to be a major mode of action in bone marrow ablative regimens.

Stepwise mechanisms consisting of initial proliferation and subsequent apoptosis can be found in other contexts in HF biology. The HFSC activation was observed in the dystrophic anagen response after ionizing radiation[7,59], and the final response to ionizing radiation was suggested to be decided according to the proliferation state of HFSCs[67]. In lichen planopilaris, which is interferon-gamma (IFNγ)-driven inflammatory permanent hair loss, HFSCs initially underwent proliferation and subsequently encountered apoptosis within their stem cell niche[38]. Considering the frequent observation of the cellular dynamics relevant in this study, a priming stem cell activation and timely induction of a successive insult is suggested to be a general mechanism that effectively leads to stem cell exhaustion

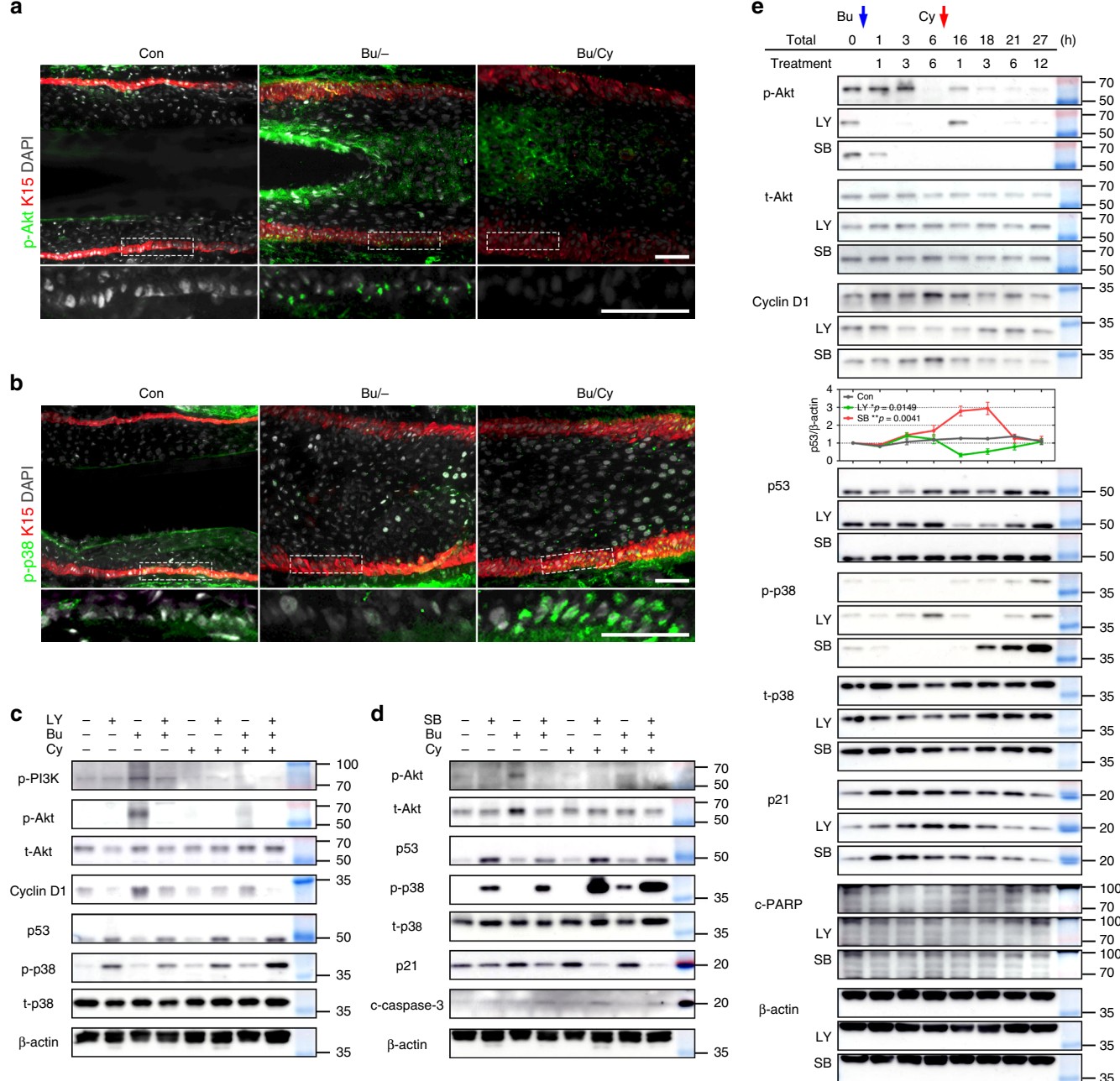

**Fig. 7** Roles of PI3K/Akt and p53/p38 in phase conversion after Bu/Cy treatment. **a** Representative images of phosphorylated Akt+ cells among K15+ HFSCs in the bulge. **b** Representative images of phosphorylated p38+ cells among K15+ HFSCs in the bulge (immunofluorescence; scale bar = 50 μm). **c** Protein analysis after Bu and/or Cy treatment with PI3K inhibitor (20 μM; LY294002) comparing the concentrations of phosphorylated PI3K, phosphorylated and total Akt, cyclin D1, p53, and phosphorylated and total p38, with β-actin as a loading control (n = 4 biological samples, Supplementary Fig. 8a). **d** Protein analysis after Bu and/or Cy treatment with p38 inhibitor (10–20 μM; SB202190 or SB203580) comparing the concentrations of phosphorylated and total Akt, p53, phosphorylated and total p38, p21, and cleaved caspase-3, with β-actin as a loading control (n = 4 biological samples, Supplementary Fig. 8b). **e** Protein analysis after sequential Bu/Cy treatment with PI3K or p38 inhibitor showing the temporal changes of phosphorylated and total Akt, cyclin D1, p53, phosphorylated and total p38, p21, and cleaved PARP, with β-actin as a loading control. Temporal quantification plots of the concentrations of p53 per β-actin (n = 4 biological samples, Supplementary Fig. 9; Western blotting). Bu busulfan, Cy cyclophosphamide, Bu/Cy busulfan followed by cyclophosphamide, LY LY294002; SB SB202190 or SB203580, c-caspase-3 cleaved caspase-3, c-PARP cleaved PARP. Data are mean ± SEM. Source data are provided as a Source Data file. *p < 0.05 (vs. Con); **p < 0.01 (vs. Con, two-way analysis of variance with Dunnett's multiple comparisons test)

not only after exposure to genotoxic agents also under inflammatory conditions[26]. Moreover, it is supposed that permanent hair loss caused by other chemotherapeutic regimens that require time-dependent combination may be driven by a similar mechanism, including mitotic catastrophe[16].

Even though the permanent CIA model in this study clarified the spatiotemporal dynamics of HFSCs after chemotherapy, the limitation is that the macroenvironment of isolated human HFs was supported by mouse skin, not by human scalp skin. Considering that human HFs operate within the complex functional

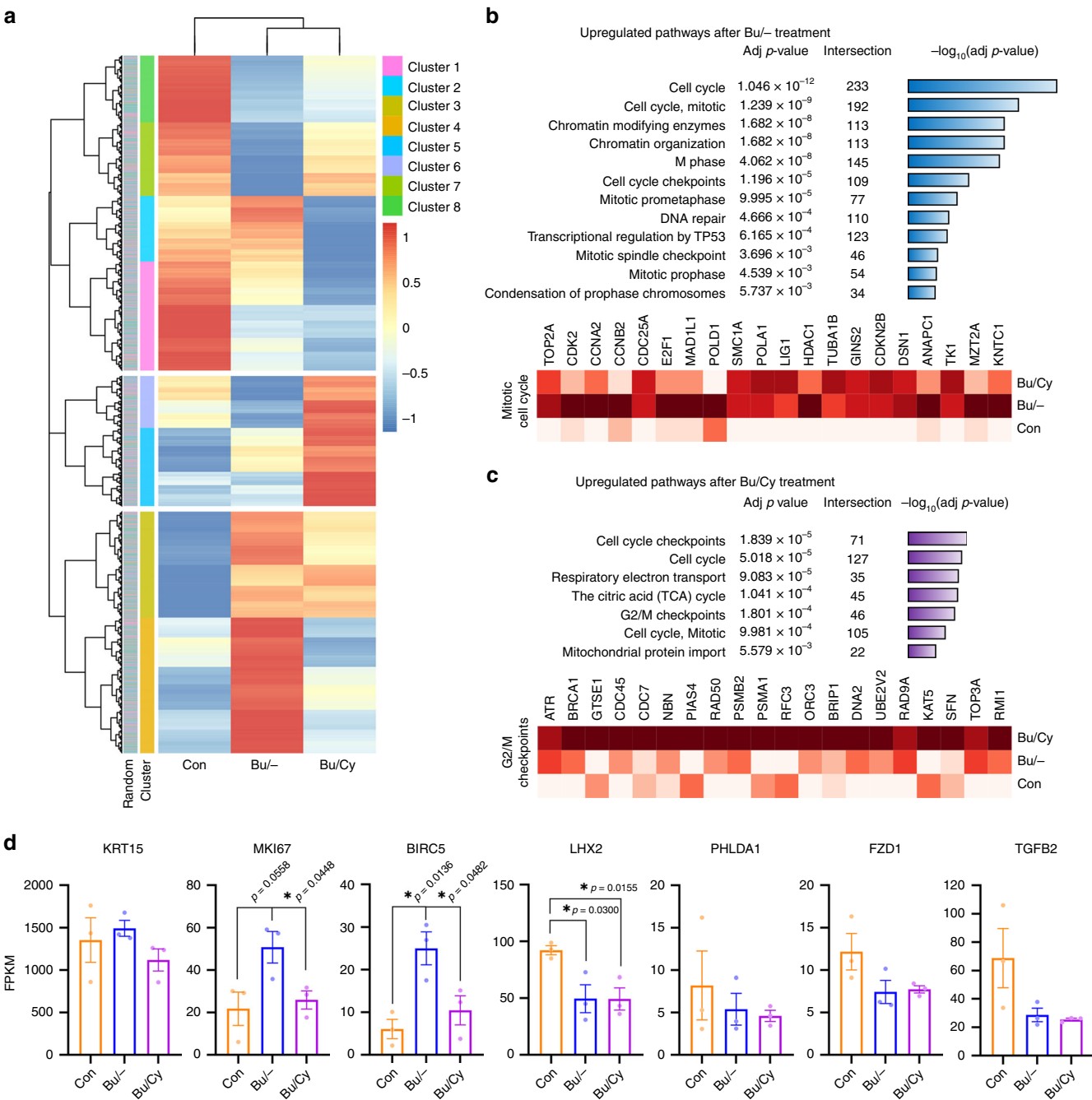

**Fig. 8** Global gene expression changes of in vivo HFSCs after Bu/Cy treatment. **a** Hierarchical clustering and heatmap analysis of the 15855 gene expressions from RNA-seq of laser capture microdissected HFSCs ($n = 3$ biological replicates/group). Based on the hierarchical clustering, the upregulated genes after Bu or Bu/Cy treatment were selected for the functional enrichment pathway analysis. **b** Selection of upregulated pathways from the Reactome database after Bu treatment (on day 4; clusters 3, 4). Expression heatmap of representative genes intersecting with the mitotic cell cycle pathway. **c** Selection of upregulated pathways from the Reactome database after Bu/Cy treatment (on day 7; clusters 5, 6). Expression heatmap of representative genes intersecting with the G2/M checkpoints pathway. **d** Expression of representative genes showing the key characteristics of human HFSCs including *KRT15* (for HFSC marker), *MKI67* (for Ki67), *BIRC5* (for Survivin), *LHX2, PHLDA1, FZD1*, and *TGFB2* (for HFSC stemness) ($n = 3$ biological replicates/group). Bu busulfan, Cy cyclophosphamide, Bu/Cy busulfan followed by cyclophosphamide; HFSC, hair follicle stem cell. Data are mean ± SEM. Source data are provided as a Source Data file. *$p < 0.05$ (unpaired *t*-test)

skin appendage unit[68], the complete elimination of HF xenografts in this model is thought to be exaggerated compared to the partial elimination of HFs in patients due to differential interfollicular susceptibility to chemotherapy.

In summary, our establishment of a permanent CIA model using human HF xenografts; comparison between transient and permanent loss models; ex vivo reproduction using HF organ culture, cell cycle analysis and signaling pathway analysis; and global gene expression profiling are uniquely poised us to explore the spatial and temporal landscape and the stepwise mechanism underlying the permanent loss of regeneration in human HFs after alkylating chemotherapy. In so doing, we provide new insights into how adult stem cells maintain their resistance to DNA damage in their niche and how they eventually lose their

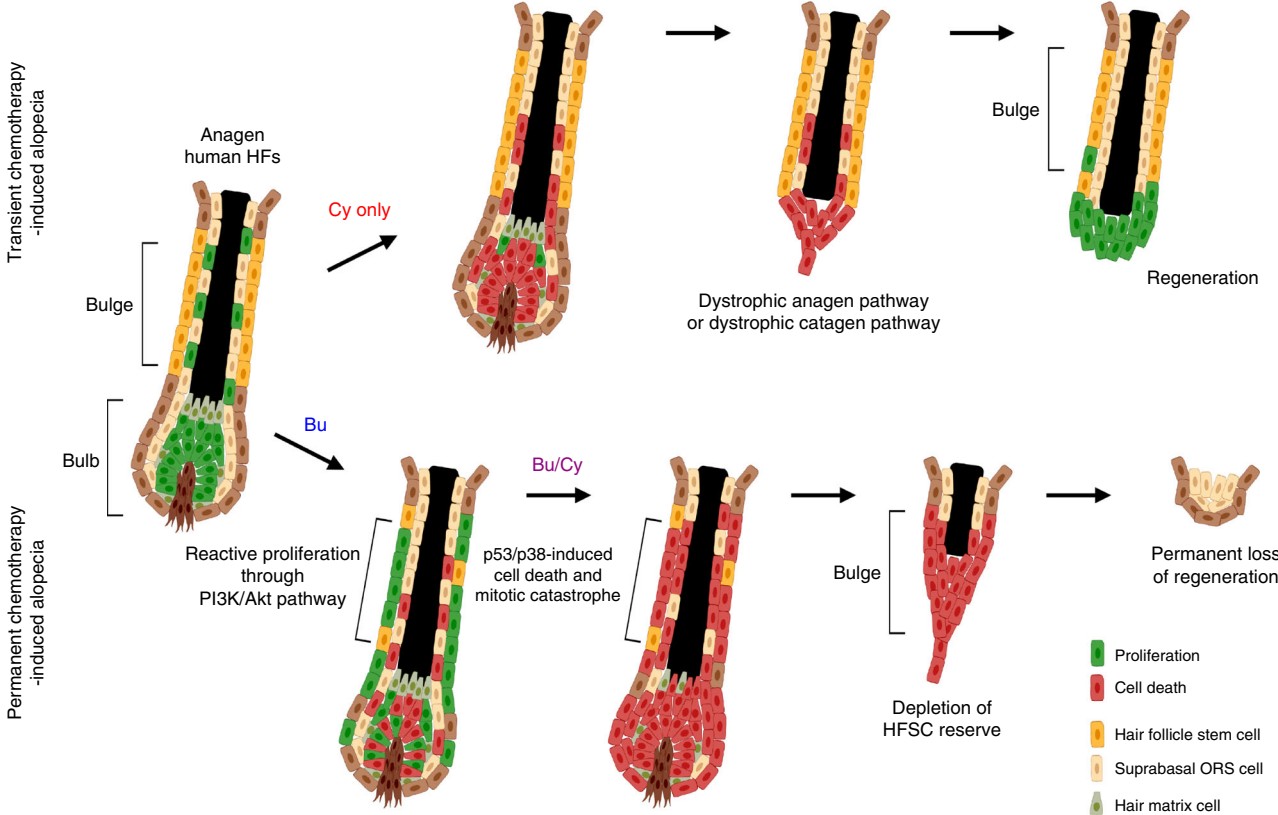

**Fig. 9** Mechanism of the permanent loss of regeneration. In transient loss condition, human HFs recover their regenerative segment based on intact HFSCs in the permanent segment through dystrophic anagen pathway or dystrophic catagen pathway[13,45]. However, in permanent loss condition, priming mobilization of HFSCs by initial Bu treatment makes quiescent HFSCs vulnerable to DNA damage-induced cell death by subsequent Cy treatment, resulting in loss of the stem cell reserve for regeneration. Bu busulfan, Cy cyclophosphamide, Bu/Cy busulfan followed by cyclophosphamide, HFSC hair follicle stem cell, ORS outer root sheath

integrity, followed by loss of the stem cell reserve and regeneration capacity after irreversible DNA damage. Although our focus here is on HF, the phenomenon we have revealed suggests that the advantages of HF as a regenerative organ will help expand our knowledge of stem cell biology to other tissues in humans.

## Methods

**Ethics statement**. This study was approved by the Institutional Review Board of the Seoul National University Hospital (approval number 1310-075-527). All human protocols conformed to the ethical principles of the Declaration of Helsinki, and written informed consent was obtained from all human subjects. All animal procedures were performed according to the American Association for the Accreditation of Laboratory Animal Care guidelines and approved by the Institutional Animal Care and Use Committee of the Seoul National University Hospital (approval number 12-0066).

**Isolation of human HFs**. Human scalp samples (2 × 1.5 cm) were obtained from the occipital scalp of 14 healthy volunteers (37–55 years old) who did not have any current or prior scalp disease. Human HFs were isolated from the human scalp tissue and divided into single follicular units with a scalpel under a dissecting stereomicroscope (Olympus, Tokyo, Japan). Microdissected human HFs were determined in the anagen stage and used for HF xenotransplantation into immunodeficient mice, HF organ culture, primary ORS cell culture and analysis.

**Treatment with alkylating agents**. For the permanent CIA model, busulfan (B2635; Sigma Aldrich, MO, USA) and cyclophosphamide (Baxter Oncology, Halle, Germany) were administered by intraperitoneal injection into mice at 22 weeks after HF xenotransplantation when transplanted human HFs were in anagen stage 6. Based on the mouse model for bone marrow transplantation, 20 mg/kg/day Bu in freshly dissolved saline was injected into the mice for 4 days, followed by the injection of 100 mg/kg/day Cy in freshly dissolved saline for 2 days. For the transient CIA model, Cy was administered by intraperitoneal injection into mice with transplanted human HFs. Based on the previously reported schedule for

dystrophic catagen pathway[13], 150 mg/kg/day Cy in freshly dissolved normal saline was injected into the mice for 1 day.

For the HF organ culture and ORS cell culture experiments, the minimal threshold concentrations were determined based on the plasma concentrations of active molecules for the conditioning regimen for HSCT. Busulfan (50 μmol/L; B2635, Sigma Aldrich) and 4-hydroperoxycyclophosphamide (40 μmol/L; sc-206885, Santa Cruz Biotechnology, Santa Cruz, CA, USA) were added to Williams' E medium for HF organ culture and to Keratinocyte Growth Medium for ORS cell culture. The roles of the PI3K/Akt pathway and p53/p38-induced cascade were investigated under treatment with specific inhibitors of PI3K (20 μM; LY294002, Sigma Aldrich) and p38 (10–20 μM; SB202190 or SB203580, Sigma Aldrich).

**Human HF xenotransplantation**. Microdissected human HFs were transplanted into 7-week-old, female, severe combined immunodeficiency hairless outbred mice (SHO mice; 18–20 g; Jackson Laboratory, ME, USA). Under general anesthesia with isoflurane, 20–28 human HFs were transplanted into the subcutaneous layer of each mouse using pen-type transplanters (OSI-10 and OSI-13; LeadM, Seoul, Korea). The grafted areas were covered with gauze and surgical adhesive, and the mice were housed in individual cages. The transplanted human HFs were regenerated into anagen stage 6 after 22 weeks ($n = 765$ HFs, 32 mice, 3 independent experiments). The mice were distributed into the following experimental groups: (1) control (Con; $n = 8$ mice), (2) transient CIA model (Cy only; $n = 7$ mice), and (3) permanent CIA model (Bu/Cy; $n = 17$ mice). For histological analysis, human HFs were harvested on day 0, day 4, day 7, day 10, day 13, and day 16. Then, HFs were collected on day 60 to investigate whether HFs would regenerate the normal anagen structure (Fig. 1a).

**Human HF organ culture**. Each HF was cut at the level of the sebaceous duct and then cultured for 6 days at 37 °C in a 5% $CO_2$ atmosphere in Williams' E medium (Gibco, Life Technologies, MA, USA) supplemented with 10 ng/ml hydrocortisone, 10 μg/ml insulin, 2 mM L-glutamine, and 1× penicillin streptomycin solution (Gibco). The elongation of the hair shaft and the morphology of the hair bulb were measured and photographed using a stereomicroscope (Olympus). HFs were scored according to their hair cycle stages at each time point (100, anagen; 200, early catagen; 300, mid-catagen; and 400, late catagen) according to the hair cycle score guidelines[28,69].

**Human ORS cell culture**. For the isolation of human ORS cells, the bulge regions of the human HFs were cut off to prevent contamination with other cells. The trimmed bulge regions were immersed in Dulbecco's modified Eagle's medium (DMEM; Welgene, Daegu, Korea) supplemented with 20% fetal bovine serum (FBS; Welgene) and 1× penicillin-streptomycin solution (Gibco) at 37 °C in a 5% $CO_2$ atmosphere. On the third day of culture, the medium was changed to KGM-Gold™ Keratinocyte Growth Medium (Lonza, MD, USA) supplemented with KGM-Gold™ SingleQuots (Lonza; hydrocortisone, transferrin, epinephrine, gentamicin, amphotericin B, bovine pituitary extract, recombinant human epidermal growth factor, and insulin) at 37 °C in a 5% $CO_2$ atmosphere.

**Histology and immunostaining**. Paraffin-embedded HFs were sectioned at a thickness of 4 μm and stained with hematoxylin and eosin. Masson Fontana staining was performed to detect melanin granules. Alkaline phosphatase activity was detected as a purple color using NBT/BCIP (Roche Diagnostics, Mannheim, Germany) substrate diluted in NTMT buffer (100 mM NaCl; 100 mM Tris-Cl; 50 mM $MgCl_2$; 0.1% Tween 20) for 20 min.

For immunohistochemistry, tissue sections were dewaxed in xylene, rehydrated using a graded alcohol series, and incubated in an endogenous peroxide-blocking solution for 5 min. Antigen retrieval was performed by incubating the sections in pH 6.0 REAL™ Target Retrieval Solution (S2031, Dako, Glostrup, Denmark) at 120 °C for 5 min. The sections were blocked with preblocking solution (GBI Labs, Bothwell, WA, USA) at 25 °C for 30 min, followed by incubation overnight at 4 °C with the following primary antibody diluted in antibody diluent reagent (Invitrogen, Carlsbad, CA, USA): p53 (SC126, 1:50; Santa Cruz Biotechnology). After three washes with PBS, the slides were incubated with secondary antibodies at 25 °C for 1 h. The antigen-antibody complex was visualized with an AEC kit (C01-12, GBI Labs) and counterstained with hematoxylin.

For immunofluorescence, frozen tissues were sectioned at a thickness of 4 μm and incubated overnight at 4 °C with the following primary antibodies diluted in antibody diluent reagent (Invitrogen): Fas (ab82419, 1:100; Abcam, MA, USA), Ki67 (SP6, 1:200; Spring Bioscience, Pleasanton, CA, USA or MIB-1, 1:200; Dako), K15 (LHK15, 1:200; Thermo Fisher), p53 (DO-7, 1:800; Novocastra, Newcastle, England or 1C12, 1:1000; Cell Signaling, Danvers, MA, USA), K14 (GTX104124, 1:1000; GeneTex, Irvine, CA, USA), cleaved caspase-3 (5A1E, 1:400; Cell Signaling), γ-H2AX (#2577, 1:800; Cell Signaling), Shh (EP1190Y, 1:500; Abcam), Lhx2 (C-20, 1:200; Santa Cruz Biotechnology), integrin β4 (H-101, 1:200; Santa Cruz Biotechnology), phospho Akt (#9275, 1:800; Cell Signaling), phospho p38 (#9211, 1:800; Cell Signaling), TYR (T311, 1:100; Thermo Fisher, Waltham, MA, USA), TRP1 (SC25543, 1:100; Santa Cruz Biotechnology), and MITF (C5, 1:100; Thermo Fisher). After three washes with PBS, the slides were incubated with secondary Alexa Fluor 488/594-labeled goat IgG antibody (1:200; Thermo Fisher) at 25 °C for 1 h. Nuclei were counterstained with 4′,6-diamidino-2-phenylindole (DAPI; Dako). For TUNEL staining, frozen tissues were stained using an ApopTag fluorescein in situ apoptosis detection kit (S7110; Merck Millipore, Darmstadt, Germany) according to the manufacturer's protocol.

Microscopic observations were performed with an Olympus microscope (Olympus) equipped with a DP70 camera and using cellSens software (Olympus) for image acquisition. Immunofluorescence observations were performed and recorded with a Leica fluorescence microscope (Leica Biosystems, Nussloch, Germany). Mx cells were defined as the lowermost portion of the HF epithelial cells below the top of the DP[70]. For quantitative analyses, the number of positive cells was counted in the Mx, ORS, and DP areas in the bulb and the basal and suprabasal layers in the bulge using ImageJ software (National Institutes of Health, Bethesda, MD, USA) with blinding regarding the experimental group.

**Flow cytometry analysis**. To ensure the cellular homogeneity of both populations, human ORS cells in the active growth and confluent quiescent stages were stained with ORS cell surface markers and analyzed by flow cytometry. ORS cells were detached after a brief treatment with 1 × 0.05% trypsin-EDTA (Gibco) and processed into single cell suspensions. Then, ORS cells were incubated with the following fluorochrome-labeled monoclonal antibodies at 4 °C for 30 min: anti-CD29 (APC-conjugated, TS2/16; BioLegend, San Diego, CA, USA), anti-CD49f (Alexa Fluor 488-conjugated, GoH3; BioLegend), anti-CD133 (PE-conjugated, 293C3; Miltenyi Biotec, Gaithersburg, MD, USA), and anti-CD200 (APC-conjugated, OX104; Invitrogen). Three-color flow cytometric analysis was performed with a FACSCalibur™ instrument (BD Biosciences, San Jose, CA, USA).

**Cell cycle analysis**. To determine the effects of the alkylating agents on the cell cycle, human ORS cells were seeded on a 100-mm plastic dish (SPL Life Sciences, Pocheon, Korea) and distributed to the following experimental groups: (1) actively growing (the cells were allowed to reach 70–80% confluence) and (2) confluent quiescent (the cells were allowed to reach 100% confluence and cultured for an additional 2 days). For the comparison of the cell cycle stages, ORS cells were treated in medium with or without Bu, Cy, or Bu plus Cy for 24 h and then harvested. For the sequential treatment of alkylating agents, ORS cells were treated in medium with or without Bu for 15 h, followed by medium with or without Cy for 24 h, and then harvested. ORS cells were trypsinized into single cell suspensions, fixed in cold 70% ethanol for 10 min and then stained with propidium iodide (PI) solution (1 μg/μL PI and 0.125% RNase A; Sigma Aldrich) at 25 °C for 15 min. Approximately 10,000 cells/sample were analyzed using a FACSCalibur™ instrument (BD Biosciences). The percentage of cells in each phase of the cell cycle was determined using ModFit LT software (BD Biosciences) and the Cell Cycle Platform in FlowJo software (FlowJo, LLC).

**Western blot analysis**. Total protein from human ORS cells was extracted using RIPA lysis buffer (Millipore, Billerica, MA, USA) according to the manufacturer's instructions. Equal amounts of total protein were separated by electrophoresis using 8–10% sodium dodecyl sulfate-polyacrylamide gels and transferred onto polyvinylidene fluoride membranes (Amersham, Buckinghamshire, UK). The blotted membranes were cut horizontally according to the size marker (Supplementary Figs. 6 and 10) and incubated with the following primary antibodies corresponding to the target protein size overnight at 4 °C: anti-phospho PI3K (#4228, 1:1000; Cell Signaling), anti-phospho Akt (#9275, 1:1000; Cell Signaling), anti-total Akt (#9272, 1:1000; Cell Signaling), anti-cyclin D1 (#2926, 1:2000; Cell Signaling), anti-p53 (#2524, 1:1000; Cell Signaling), anti-phospho p38 (#9211, 1:1000; Cell Signaling), anti-total p38 (#9212, 1:1000; Cell Signaling), anti-p21 (#2947, 1:1000; Cell Signaling), anti-cleaved caspase-3 (#9664, 1:1000; Cell Signaling), anti-PARP (#9542, 1:1000; Cell Signaling) and anti-β-actin (SC-1616, 1:1000; Santa Cruz Biotechnology or BA3R, 1:5000; Thermo Fisher). The membranes were probed with anti-rabbit IgG and anti-mouse IgG antibodies (horseradish peroxidase-conjugated, GTX213110, GTX213111, 1:5000; GeneTex, Irvine, CA, USA) at 25 °C for 1 h. The blotted membranes were restored by removing bound primary and secondary antibodies using Restore™ PLUS Western Blot Stripping Buffer (Thermo Fisher) and reprobed for the different size target protein compared to the original target protein. Antibody-antigen complexes were visualized using an enhanced chemiluminescence system (Thermo Scientific) and captured by an Amersham Imager 680 (GE Healthcare, Chicago, IL, USA). The captured digital images were quantified and analyzed using Amersham Imager 680 Analysis Software (GE Healthcare).

**Laser capture microdissection**. To preserve RNA integrity, a microtome cutting and cresyl violet staining protocol was used for frozen sections according to the manufacturer's instructions. Serial longitudinal cryosections were generated from the in vivo HF xenografts on MembraneSlides (Carl Zeiss, Munich, Germany), and the bulge area was identified as previously described[38,43]. Using a PALM MicroBeam (Carl Zeiss), a focused laser beam catapulted the basal cells in the bulge regions into an AdhesiveCap 500 (Carl Zeiss) positioned above the section (×20 lens, Power 55%; Supplementary Fig. 11). A total of 300–500 HFSCs were collected for each replicate ($n = 3$ biological replicates/group) and stored at −80 °C before RNA library preparation.

**RNA-seq library preparation**. Total RNA was isolated and purified from the collected cells with a NucleoSpin RNA XS kit (740902, TaKaRa Bio, Ostu, Japan). The quality of purified total RNA was evaluated based on the RNA Integrity Number (RIN) calculated with an Agilent 2100 Bioanalyzer with an Agilent RNA 6000 Pico kit (5067-1513, Agilent Technologies, Waldbronn, Germany). The RNA-seq library was prepared using SMARTer Stranded Total RNA-Seq Kit v2 (Pico Input Mammalian, 634412, TaKaRa Bio). The generated library was validated using an Agilent 2100 Bioanalyzer with an Agilent High Sensitivity DNA Kit (5067-4626, Agilent Technologies).

**RNA-seq analysis**. The libraries were subjected to paired-end sequencing with a 100-base pair length on an Illumina HiSeq 2500 sequencer (Illumina). The sequenced paired-end reads were aligned to the human reference genome (hg38), and fragments per kilobase per million reads (FPKM) for each gene were calculated from the aligned reads using RSEM-1.3.0. For the hierarchical clustering, a mean expression matrix was calculated by averaging the FPKM values of the replicates of each group. Then, the FPKM matrix was log2 transformed and normalized among the groups, and the heatmap was subsequently generated using the R package pheatmap. Based on the cluster list of upregulated genes, functional enrichment pathway analyses were performed using the g:Profiler online tool. The upregulated pathway networks were created based on the Reactome database and visualized using Cytoscape software.

**Statistical analysis**. Statistical significance was determined with Student's $t$-test, one-way analysis of variance with a post hoc test, or two-way analysis of variance with a post hoc test. All tests were two-tailed, and differences with $p < 0.05$ were considered statistically significant.

**Reporting summary**. Further information on research design is available in the Nature Research Reporting Summary linked to this article.

## Data availability

The accession numbers for the RNA-seq data reported in this paper are available in the NCBI Gene Expression Omnibus (GEO) archive (accession GEO GSE130054). The source data underlying Figs. 2, 3, 4, 5, 7, 8, Supplementary Figs. 1, 7, 8, and 9 are provided as a Source Data file. All supplementary materials and a reporting summary for this article are available in the Supporting Information file.

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

## Acknowledgements

This work was supported by a National Research Foundation of Korea (NRF) grant funded by the Korean government (NRF-2017R1A2B4006212).

## Author contributions

J.Y.K. developed experimental models, performed experiments, interpreted the data, analyzed RNA-seq data, and wrote the paper. J.O. contributed to in vivo experiments and wrote the paper. J-S.Y. contributed to in vivo experiments. B.M.K. and M.P. contributed to in vitro experiments. S.K., W.L. and J-I.K. contributed to RNA-seq analysis. S.H. and K.H.K. discussed and interpreted the data. O.K. supervised the project, developed experimental models, interpreted the data, and wrote the paper. All authors approved the paper.

## Additional information

**Competing interests:** The authors declare no competing interests.

