## [Peer Review File · Nature Communications]

Reviewers' comments:

Reviewer #1 (Remarks to the Author):

Permanent chemotherapy-induced alopecia (PCIA)/hair loss is a clinically important but rarely explored disease. In cancer patients undergoing chemotherapy or combination of chemotherapy and radiotherapy, up to 10~20% of the patients can be affected. The severity of this disease can vary from mild alopecia to very severe alopecia. Histologically, this type of alopecia is characterized by loss of hair follicles, consistent with cicatricial alopecia. Due to permanent loss hair follicles, there is no effective treatment. How the hair follicles are permanently lost currently is unknown. The lack of mechanistic exploration is due to, at least in part, a lack of an experimental model. Establishment of an experimental model for PCIA and characterization of the cell dynamics and molecular mechanism are of high clinical and biological significance. The principles learned can be of great potential to be applied to the research of chemotherapy injury to other organs.

To tackle the puzzle of PCIA, on this manuscript, Kim et al. developed a humanized model of PCIA. They transplanted human hair follicles to immunocompromised mice and waited until the hair follicles entered anagen to mimic human scalp hair growth. When the hair follicles are actively growing, the mice were treated with busulfan, followed by cyclophosphamide. This regimen mimics the clinical regimen that frequently induces PCIA. They showed that, following busulfan treatment, bulge stem cells were activated in a dystrophic anagen response. It has been shown that quiescent bulge stem cells are resistant to both chemotherapy and radiotherapy. However, during their reactive activation induced by busulfan, bulge stem cells became susceptible to the treatment of cyclophosphamide, undergoing extensive apoptosis. The extensive loss of bulge stem cells was associated with permanent loss of hair follicles. Mechanistically, they showed that busulfan might activate bulge stem cells through PI3K/Akt pathway and bulge stem cell death might be mediated by p38-dependent cell death. The molecular mechanisms revealed here might seem a bit preliminary. However, considering the limited availability of human hair follicle samples, this work might shed new light on the understanding of this disease. Overall, the model they set up here can pay a new way for future investigation of PCIA and this work can potentially provide new insight into the cellular/molecular basis of PCIA.

I have some suggestion for the authors.

Major points

1. Figure 1 and Figure 2 showed humanized mouse model to characterize hair follicle loss and cell dynamics. Figure 3 showed loss of melanocytes in humanized mouse model. Figure 7 showed humanized mouse model to demonstrate apoptosis of bulge stem cell after Bu/Cy. I would suggest the authors to move Figure 7 to Figure 3 and this will help to highlight how bulge stem cells are lost. Figure 3 seems a bit redundant; moving this figure to supplementary data might be a good choice. Figure 4 showed ex vivo hair follicle organ culture result. I wonder how this figure can help here. I would suggest the authors to incorporate Figure 4 to Figure 7 if the results need to be shown.
2. In Figure 2, the last dose of Cy was administered on Day 5. However, there are still extensive p53+ cells, Fas+ cells and TUNEL+ cells in ORS on Day 10. How could the effect of Cy persist for 5 days? Did the authors detect persistent DNA double strand break in ORS and bulge stem cells at this time point (for example, gamma-H2AX)?
3. In Figure 7C, the staining for p53 is shown in a relatively lower power view. Whether the cells are in the basal or suprabasal layer can not be clearly visualized. I would suggest the authors to co-stain basal cell markers (using K5 or K14 or bulge stem cell markers such as K15, CD200 or K19) and show the result in a higher power. In addition, a quantitative analysis of p53 expression should be provided. In addition, a direct demonstration of apoptosis of bulge stem cells in the humanized mouse model (such as activated caspase 3 and TUNEL) should be provided. If this is not observed in the humanized mouse model (authors showed cleaved caspase in explant culture in Figure 4), authors should explain it. How is the loss of bulge stem cells induced by Bu/Cy in vivo? Did authors detect DNA damage in bulge stem cells following the Cy treatment? In Figure 7d, a higher power view can also be helpful to demonstrate the results.
4. In Figure 7b, a quantitative analysis of Ki67 expression will be helpful. It seems that Ki67+ cells (white arrows) in Bu/- group are on the mesenchymal side of the hair follicle, or the authors rotated

the enlarged boxed area? In addition, the Ki67+ cells seem to be negative for K15 staining. Can the authors comment on this?

5. In Figure 6, authors showed the alterations of Akt/PI3K/cyclin D1 and p53/p38/p21/Bax in holoclone-rich ORS cells treated by Bu, Cy Bu/Cy. Is the protein phosphorylation or protein expression also detected in the humanized mouse model (such as immunostaining)? Additionally, authors are suggested to test the pathways by perturbing them in vitro (such as inhibition of Akt/PI3K by small molecules) and see whether cell proliferation and cell apoptosis are inhibited.

6. In reference 21, it was suggested the response of hair follicles to ionizing radiation can vary according to the activation states of hair follicle stem cells. Authors are suggested to discuss this to support their observation. In reference 11, the activation of bulge stem cells in dystrophic anagen response was observed radiotherapy. Since radiotherapy and chemotherapy are both genotoxic, authors are suggested to discuss whether the activation of bulge stem cells is a general phenomenon following either chemotherapy and radiotherapy. The similar mechanism might also lead to permanent alopecia following radiotherapy.

Minor points

1. In Figure 1c, day 60, is the sebaceous gland still preserved? Did sebaceous gland cell also undergo apoptosis after Bu/Cy treatment? If not, can the authors comment on this?

2. In Figure 1c and 1e, dermal papilla is lost after Bu/Cy treatment. Dermal papilla cells are resistant to chemotherapy and radiotherapy (Reference 11, 22, 66 in the manuscript). In Figure 2B and 2C, it seems that Fas was induced in DP by Bu/Cy. TUNEL seemed to be negative in dermal papilla (Day 4). Did authors detect apoptosis of dermal papilla cells? If not, authors are suggested comment on this.

3. In Figure 2C, there are limited TUNEL+ cells in the regressing hair bulb. Can this explain how the hair bulb continue to regress?

4. In Figure 2, how did authors define "matrix cells" for quantification? Did they count all the epithelial cells below the top of dermal papilla?

5. In Figure 3, authors used TYR and TRP1 staining to detect melanocytes. These two enzymes are markers for differentiated melanocytes. Other markers (such as Mitf, TRP2, etc.) might help to demonstrate whether undifferentiated melanocytes are still present there. In addition, did the melanocytes undergo apoptosis?

6. In Figure 4d, in vitro testing showed that Bu treatment showed dystrophic changes of hair bulb, but Bu/Cy treatment did not lead to hair bulb dystrophy. Could the authors examine the histology and cell proliferation/apoptosis of the hair bulbs? This can help to verify whether the in vitro model is consistent with the humanized mouse model.

7. In Figure S1, "Bu-induced proliferation in the basal layer in the bulb area". In this figure, the indicated proliferating cells (white arrows) are mostly in the ORS above the hair bulb. It is not consistent with the title of this figure.

8. In Figure S4. The cells do not seem to be confluent in culture. If authors meant to culture holoclones-rich ORS cells into confluency and test the effect of Bu, Cy, Bu/Cy, how was 100% confluency determined?

9. There are a couple of grammatical errors in the manuscript. Authors are suggested to go over their manuscript carefully to correct these errors.

Reviewer #2 (Remarks to the Author):

The topic studied here is of great clinical relevance as there is an increasing number of cases where alopecia after chemotherapy (CIA) is permanent, suggesting major, irreversible stem cell damage. Yet, the latter has been poorly investigated. Therefore, the authors are to be commended for tackling this major unsolved problem in clinical oncology by using the instructive & clinically relevant humanized mouse CIA model they had developed & published before (JID 2016).

That damaged human bulge stem cells initially undergo (under inflammatory, IFN γ -driven conditions)

proliferation and subsequently are driven into apoptosis within their stem cell niche has already been documented for another form of human permanent alopecia, lichen planopilaris (Harries et al. J Pathol 2013). This has invited the "stem cell exhaust" hypothesis in the pathobiology of permanent alopecia (Harries et al. Trends Mol Med 2018). The same phenomenon is seen in the rapidly proliferating progeny of human bulge stem cells, i.e. in the transit amplifying cells of the anagen hair bulb under conditions of chemotherapy in organ-cultured human scalp HFs (Bodo et al. AJP 2007). Surprisingly, neither of these previously reported observations, which are directly relevant in the current context, are properly cited and discussed here.

That chemotherapy can induce permanent alopecia with the morphological correlate of HF deletion (incl. loss of the so-called permanent part of the HF) is well-known and has been described in many clinical and limited dermatopathological reports and is thus not novel either. The same applies to the finding that the cellular response to alkylating agents is highly cell cycle status-dependent. In any case, the data provided here regarding the proliferation-dependence of human HF stem cell responses to alkylating agents are only correlative, and attempts to selectively arrest the HF stem cells in defined cell cycle phases so as to observe how this differentially impacts on their chemotherapy response were not made.

Conceptually, this appears to restrict the novelty of the findings reported here to the observation that HFSC proliferation presumably was activated through the PI3K/Akt pathway, and that depletion may have been driven by p38-dependent cell death. This is interesting since CIA-associated apoptosis, at least in the hair matrix, is widely believed to be p53-dependent (Botchkarev et al. AJP 2000). However, this dogma has previously been challenged by findings in chemotherapy-treated feather follicles, where the dominant molecular matrix keratinocyte response is the down-regulation of Shh, not p53-dependent apoptosis (Xie et al. JID 2015) - another study that the authors may wish to consult and discuss in the context of their findings. Regrettably, the role of Shh in the events leading up to human bulge epithelial stem cell apoptosis was not investigated.

In any case, the postulated role of PI3K/Akt in stem cell proliferation, and of p38 in stem cell apoptosis is only assumed on the basis of correlative data, but not definitively proven. Global gene expression profiling of the human bulge in response to chemotherapy during different time points, using laser capture microdissection, which has been used before when investigating human permanent alopecia (Harries et al. 2013, Imanishi et al. JID 2018), would be an excellent method for interrogating the molecular damage response pathways of bulge stem cells in situ much more instructively and comprehensively. Also, a number of techniques have been published that permit one to isolate or at least enrich for human bulge-derived epithelial stem cells. Using one of these human HF stem cell culture methods, mechanistic studies would seem possible through which the authors' PI3K/Akt and p38 hypotheses could likely be verified.

Finally, have the authors considered how one - potentially important - methodological weakness of their CIA model, namely the transplantation of isolated scalp HFs rather than of full-tickness human scalp skin - might have impacted on the results they obtained? There is increasing appreciation that human HFs operate in the context of a complex functional skin appendage unit (i.e. HF+ arrector pili muscle + sebaceous gland + eccrine gland coil + a cone of dermal adipocytes that enwraps all of these structures, incl. the bulge [Poblet et al. BJD 2018]). Since all of these human tissues/cells are missing in their hair xenotransplant model (Yoon et al. JID 2016), one wonders to which extent the absence of this distinct human peri-bulge tissue signaling milieu may have exaggerated, disrupted or distorted the HF stem cell responses to chemotherapy observed here. It would seem appropriate to at least discuss this possibility.

Reviewer #3 (Remarks to the Author):

This is a very interesting article and deserves publishing.

1. I would like to see in discussion speculations on other classes of drugs that may follow similar mechanism of effect in hair follicle.
2. Please expand and comment on the importance of timing of successive insult by the chemotherapeutic agents for permanency of alopecia - in humans.

We are pleased to submit a revised version of our manuscript entitled "Priming mobilization of hair follicle stem cells triggers permanent loss of regeneration after alkylating chemotherapy" for consideration for publication in Nature Communications. We appreciate the referees for their review of our manuscript and their constructive feedback. We have carefully examined the reviewers' comments and feel confident that we have addressed the major concerns, including 1) direct evidence of bulge stem cell apoptosis and characterization of DNA damage, 2) mechanistic molecular data establishing the role of the p38 and PI3K/Akt pathways, and 3) the response of hair follicle stem cells arrested at defined cell cycle stages, including global gene expression changes, through additional experiments.

Reviewers' comments:

Reviewer #1 (Remarks to the Author):

Permanent chemotherapy-induced alopecia (PCIA)/hair loss is a clinically important but rarely explored disease. In cancer patients undergoing chemotherapy or combination of chemotherapy and radiotherapy, up to 10~20% of the patients can be affected. The severity of this disease can vary from mild alopecia to very severe alopecia. Histologically, this type of alopecia is characterized by loss of hair follicles, consistent with cicatricial alopecia. Due to permanent loss hair follicles, there is no effective treatment. How the hair follicles are permanently lost currently is unknown. The lack of mechanistic exploration is due to, at least in part, a lack of an experimental model. Establishment of an experimental model for PCIA and characterization of the cell dynamics and molecular mechanism are of high clinical and biological significance. The principles learned can be of great potential to be applied to the research of chemotherapy injury to other organs.

To tackle the puzzle of PCIA, on this manuscript, Kim et al. developed a humanized model of PCIA. They transplanted human hair follicles to immunocompromised mice and waited until the hair follicles entered anagen to mimic human scalp hair growth. When the hair follicles are actively growing, the mice were treated with busulfan, followed by cyclophosphamide. This regimen mimics the clinical regimen that frequently induces PCIA. They showed that, following busulfan treatment, bulge stem cells were activated in a dystrophic anagen response. It has been shown that quiescent bulge stem cells are resistant to both chemotherapy and radiotherapy. However, during their reactive activation induced by busulfan, bulge stem cells became susceptible to the treatment of cyclophosphamide, undergoing extensive apoptosis.

The extensive loss of bulge stem cells was associated with permanent loss of hair follicles. Mechanistically, they showed that busulfan might activate bulge stem cells through PI3K/Akt pathway and bulge stem cell death might be mediated by p38-dependent cell death. The molecular mechanisms revealed here might seem a bit preliminary. However, considering the limited availability of human hair follicle samples, this work might shed new light on the understanding of this disease. Overall, the model they set up here can pay a new way for future investigation of PCIA and this work can potentially provide new insight into the cellular/molecular basis of PCIA.

I have some suggestion for the authors.

Major points

1. Figure 1 and Figure 2 showed humanized mouse model to characterize hair follicle loss and cell dynamics. Figure 3 showed loss of melanocytes in humanized mouse model. Figure 7 showed humanized mouse model to demonstrate apoptosis of bulge stem cell after Bu/Cy. I would suggest the authors to move Figure 7 to Figure 3 and this will help to highlight how bulge stem cells are lost. Figure 3 seems a bit redundant; moving this figure to supplementary data might be a good choice. Figure 4 showed ex vivo hair follicle organ culture result. I wonder how this figure can help here. I would suggest the authors to incorporate Figure 4 to Figure 7 if the results need to be shown.

Response: We agree with the reviewer, and we rearranged the order of the figures to show the phenotype when human HFSCs are lost in the humanized mouse model in the first section and then focused on the molecular mechanism and global gene expression changes in the subsequent section. The original Figure 3 showing the loss of melanocytes is now moved to Supplementary Figure 1. We agree that the ex vivo reproduction of HFSC loss in HF organ culture may not be an integral part of this study, but we respectfully decided not to remove it or move it to the supplementary data yet. These results could provide future readers with supportive evidence of HFSC loss in the solid and classical disease model using human HFs. More importantly, based on HF organ culture, we established an experimental basis for the dose of chemotherapeutic agents used in further experiments. We considered incorporating Figure 3 into Figure 4 (in the revised manuscript), but it seemed better to leave these data separate to deliver accurate messages to readers. If it is still necessary after this revision, we will be willing to merge the two figures into a single figure or move these data to the

supplementary data. Thank you for your constructive input.

2. In Figure 2, the last dose of Cy was administered on Day 5. However, there are still extensive p53+ cells, Fas+ cells and TUNEL+ cells in ORS on Day 10. How could the effect of Cy persist for 5 days? Did the authors detect persistent DNA double strand break in ORS and bulge stem cells at this time point (for example, gamma-H2AX)?

Response: We appreciate this insightful and important comment. According to your input, we assessed the expression of γ -H2AX to determine whether DNA double-strand breaks persist in ORS and bulge stem cells in a comparison between the transient and permanent models (Figure 3e). After Cy only treatment in the transient model, H2AX phosphorylation occurred mostly in the suprabasal ORS cells, and double-strand breaks seemed to be resolved as time passed. However, γ -H2AX⁺ cells were detected massively in the ORS and bulge stem cells after Bu/Cy treatment and did not disappear until day 10, indicating the persistence of unrepaired DNA damage. Based on this finding, we can show the type and extent of actual DNA damage in human HF and the increased susceptibility to Cy treatment after priming Bu treatment. Thank you so much for your constructive insight.

3. In Figure 7C, the staining for p53 is shown in a relatively lower power view. Whether the cells are in the basal or suprabasal layer cannot be clearly visualized. I would suggest the authors to co-stain basal cell markers (using K5 or K14 or bulge stem cell markers such as K15, CD200 or K19) and show the result in a higher power. In addition, a quantitative analysis of p53 expression should be provided. In addition, a direct demonstration of apoptosis of bulge stem cells in the humanized mouse model (such as activated caspase 3 and TUNEL) should be provided. If this is not observed in the humanized mouse model (authors showed cleaved caspase in explant culture in Figure 4), authors should explain it. How is the loss of bulge stem cells induced by Bu/Cy in vivo? Did authors detect DNA damage in bulge stem cells following the Cy treatment? In Figure 7d, a higher power view can also be helpful to demonstrate the results.

Response: We agree that the evidence of bulge stem cell apoptosis should be presented more directly. According to your suggestion, we performed additional experiments and revised the figure with a high-power view and the corresponding quantification showing how the p53⁺ cell population changed in the K14⁺ basal and suprabasal bulge of in vivo HF xenografts in the

transient and permanent loss models (Figure 3c). We also considered K15 and K19 ^{Exp Dermatol. 2015 Jun;24(6):462-7} as markers of bulge stem cells, but the host of both antibodies is the same as that of the current p53 antibody. To show direct evidence of bulge stem cell apoptosis, we performed experiments showing cleaved caspase-3⁺K15⁺ cells in the bulge of in vivo HF xenografts in the permanent loss model (Figure 3d). Basal bulge cells underwent extensive apoptosis through the activation of caspase-3 in the K15⁺ basal layer after Bu/Cy treatment. We added an enlarged section for each figure after Lhx2 staining (Figure 3g), and the pattern of Lhx2 expression in human HFs was obtained from a previous human HF study ^{Exp Dermatol. 2015 Jun;24(6):462-7}. Thank you for this constructive comment.

4. In Figure 7b, a quantitative analysis of Ki67 expression will be helpful. It seems that Ki67⁺ cells (white arrows) in Bu/- group are on the mesenchymal side of the hair follicle, or the authors rotated the enlarged boxed area? In addition, the Ki67⁺ cells seem to be negative for K15 staining. Can the authors comment on this?

Response: We regret that we included a misleading section about the Bu/- group in Figure 7b (now Figure 3b). This section has been replaced with another section with clearer details. We added a quantitative analysis of Ki67⁺K15⁺ cells comparing the transient and permanent loss models.

5. In Figure 6, authors showed the alterations of Akt/PI3K/cyclin D1 and p53/p38/p21/Bax in holoclone-rich ORS cells treated by Bu, Cy Bu/Cy. Is the protein phosphorylation or protein expression also detected in the humanized mouse model (such as immunostaining)? Additionally, authors are suggested to test the pathways by perturbing them in vitro (such as inhibition of Akt/PI3K by small molecules) and see whether cell proliferation and cell apoptosis are inhibited.

Response: We are grateful for this insightful comment. According to the reviewers' and editor's comments, we performed additional experiments to determine the molecular mechanism regarding the role of the PI3K/Akt and p53/p38 pathways and global gene expression changes in HFSCs after Bu/Cy treatment, as follows:

To determine the temporal dynamics after sequential Bu/Cy treatment, ORS cells were consecutively harvested after 0, 1, 3, and 6 h of priming Bu treatment and then after 1, 3, 6,

and 12 h of subsequent Cy treatment (Figure 6b). Protein analysis showed dramatic changes in two different phases: the PI3K/Akt pathway activation with upregulated cyclin D1 after Bu treatment, and the subsequent conversion into p53/p38-induced cell death with PI3K/Akt pathway inhibition after Bu/Cy treatment, resulting in dynamic changes in p21, representing the cellular decision of cell survival or cell death (Figure 6d).

To confirm the phase conversion of the DNA damage response, we performed immunostaining for the phosphorylated Akt and p38 proteins in the *in vivo* HF xenografts. Consistent with the protein analysis, Akt phosphorylation occurred after Bu treatment (Figure 7a), and p38 phosphorylation occurred after Bu/Cy treatment (Figure 7b) in the K15+ HFSCs.

Then, we investigated the roles of the PI3K/Akt pathway and the p38-induced cascade with specific inhibitors of PI3K (20 μ M; LY294002) and p38 (10-20 μ M; SB202190 or SB203580). Under treatment with the PI3K inhibitor, Akt phosphorylation was completely blocked, with loss of cyclin D1 induction (Figure 7c). Notably, p53 and phosphorylated p38 were upregulated during blockade of the PI3K/Akt pathway, supporting the hypothesis that the PI3K/Akt pathway is the active part of the DNA repair process. Unexpectedly, the driving force of p38 phosphorylation after alkylating chemotherapy was too strong to be blocked by treatment with p38 inhibitors (Figure 7d). Overwhelming potentiation of p38 phosphorylation was observed, with a DNA damage severity-dependent downregulation of p21 and a marginal increase in cleaved caspase-3.

The roles of the PI3K/Akt pathway and p53/p38-induced cascade were clearly observed during phase conversion after sequential Bu/Cy treatment (Figure 7e). The temporal changes in p53 combined with other proteins directly demonstrated two points: the first is that DNA damage sensitivity remains low if there is no priming proliferation by a PI3K inhibitor, and the other is that cells with DNA damage are not efficiently eliminated by insufficient p38 activation even at the last timepoint.

To consolidate the HFSC response in permanent CIA in humans, global gene expression changes were evaluated in *in vivo* HF xenografts after sequential Bu/Cy treatment. Using laser capture microdissection, HFSCs were obtained from the basal bulge layer and analyzed with RNA-seq (Figure S6). Based on hierarchical clustering (Figure 8a), the upregulated genes after Bu or Bu/Cy treatment were selected for functional enrichment pathway analysis from the Reactome database. Mitotic cell cycle pathways and responsible genes were upregulated after Bu, accompanied by the DNA repair pathway and p53 transcriptional regulation (Figure 8b and S7a). Intriguingly, cell cycle checkpoint pathways, especially G2/M checkpoints and

responsible genes, were upregulated after Bu/Cy treatment, along with mitochondrial membrane-associated pathways (Figure 8c and S7b).

This result indicates that the majority of HFSCs experienced mitotic catastrophe and stalled in S phase with activation of the G2/M checkpoint, which is consistent with the S phase arrest observed during the ORS cell response according to proliferating status ^{Nat Rev Cancer 7, 861-869 (2007)}. ^{Oncogene 23, 2825-2837 (2004)}. Gene expression of Ki67 (MKI67) and Survivin (BIRC5), an apoptosis inhibitor, was upregulated after Bu treatment and subsequently decreased after Bu/Cy treatment (Figure 8d). Genes for maintaining HFSC stemness (LHX2, PHLDA1, FZD1, and TGFB2) were already downregulated after Bu treatment, supporting the role of priming proliferation in loss of the stem cell reserve.

p53-dependent apoptosis is widely believed to be an essential player for the preservation of genomic integrity in response to DNA damage in the HF epithelium. In addition, p38 kinase activation is induced by DNA cross-linking agents and is sustained for more than a few days, triggering the apoptotic cascade ^{Mol Biol Cell 14, 2071-2087 (2003)}. ^{Trends Mol Med 12, 440-450 (2006)}. p38-induced apoptosis can be driven not only in a p53-dependent manner ^{EMBO J 18, 6845-6854 (1999)} but also by a p53-compromised state that originates from the overwhelming extent of DNA damage ^{Mutat Res 836, 89-97 (2018)}. More interestingly, initiation of the G2/M checkpoint requires the activation of p38 kinase, and this p38-mediated G2/M checkpoint also occurs in p53-deficient cells ^{Nature 411, 102-107 (2001)}. In this study, it is supposed that p53 activation is partially responsible for the apoptotic loss of HFSCs during the early period after DNA damage, but sustained p38 activation augmented the resulting apoptosis as a major mediator of mitotic catastrophe after irreversible DNA damage.

6. In reference 21, it was suggested the response of hair follicles to ionizing radiation can vary according to the activation states of hair follicle stem cells. Authors are suggested to discuss this to support their observation. In reference 11, the activation of bulge stem cells in dystrophic anagen response was observed radiotherapy. Since radiotherapy and chemotherapy are both genotoxic, authors are suggested to discuss whether the activation of bulge stem cells is a general phenomenon following either chemotherapy and radiotherapy. The similar mechanism might also lead to permanent alopecia following radiotherapy.

Response: We appreciate this constructive comment. According to the reviewer's comments, we added a suggestion about a generalized mechanism for permanent HFSC loss as follows:

“Stepwise mechanisms consisting of initial proliferation and subsequent apoptosis can be found in other contexts in HF biology. The activation of HFSCs was also observed in the dystrophic anagen response after radiotherapy, and the final response to ionizing radiation was suggested to be decided according to the activation state of HFSCs. In lichen planopilaris, which is interferon-gamma (IFN γ)-driven inflammatory permanent hair loss, HFSCs initially underwent proliferation and subsequently underwent apoptosis within their stem cell niche. Considering the frequent observation of the cellular dynamics relevant in this study, a priming stem cell activation and timely induction of a successive insult is suggested to be a general mechanism that effectively leads to stem cell exhaustion not only after exposure to genotoxic agents also under inflammatory conditions. Moreover, it is supposed that permanent hair loss caused by other chemotherapeutic regimens that require time-dependent combination may be driven by a similar mechanism, including mitotic catastrophe.”

Minor points

1. In Figure 1c, day 60, is the sebaceous gland still preserved? Did sebaceous gland cell also undergo apoptosis after Bu/Cy treatment? If not, can the authors comment on this?

Response: In fact, it was very difficult to find a distinguishable remnant of human HFs or tissues on day 60. Based on this, we cannot make a concrete conclusion about the sebaceous gland, but we could not find any evidence for the survival of sebaceous glands.

2. In Figure 1c and 1e, dermal papilla is lost after Bu/Cy treatment. Dermal papilla cells are resistant to chemotherapy and radiotherapy (Reference 11, 22, 66 in the manuscript). In Figure 2B and 2C, it seems that Fas was induced in DP by Bu/Cy. TUNEL seemed to be negative in dermal papilla (Day 4). Did authors detect apoptosis of dermal papilla cells? If not, authors are suggested comment on this.

Response: We identified apoptotic features, such as Fas⁺ cells and few TUNEL⁺ cells, in the DP area. It was very hard to define a strict boundary for the DP structure because the DP shrank and disappeared after Bu/Cy treatment (day 7). DP cells are known to be a very quiescent population J Cell Sci 2011 124: 1179-1182. It can be suggested that DP cells have intrinsically high resistance to chemotherapy or radiotherapy, and we could not detect evidence for the reactive proliferation of DP cells after Bu treatment. Therefore, we speculate that few DP cells

undergo apoptosis after Bu/Cy treatment, but some DP cells may escape from the surrounding epithelium after Bu treatment, as in the catagen transition.

3. In Figure 2C, there are limited TUNEL+ cells in the regressing hair bulb. Can this explain how the hair bulb continue to regress?

Response: We agree with the reviewer because we also expected that there should be more TUNEL+ apoptotic cells in the bulb area. However, if we look at a previous study ^{Am J Pathol. 2007 Oct;171(4):1153-67} that showed how the human HF bulb responds to Cy, there is a relatively small number of p53+ cells in the HF epithelium in the physiological range of 4-HC (30 µM). If we increase the dose of Cy higher than the physiological range, it should induce more apoptosis with more TUNEL+ cells in the HF epithelium. However, this process cannot be the mechanism of permanent CIA in humans because it not only depletes HFs but also causes treatment-related mortality. For the normal catagen transition, TUNEL+ cells are usually not detected in the bulb area as much as we would expect, according to a previous study ^{J Invest Dermatol. 2016 Jan;136(1):34-44}. Thank you for this suggestion.

4. In Figure 2, how did authors define “matrix cells” for quantification? Did they count all the epithelial cells below the top of dermal papilla?

Response: We defined the matrix cells as the epithelial cells below the top of the DP ^{Physiol Rev 81: 449–494 (2001), Int J Dermatol. 2014 Mar;53(3):331-41}. We added a description in the Methods section as follows: “Mx cells were defined as the lowermost portion of the HF epithelial cells below the top of the DP. For quantitative analyses, the number of positive cells was counted in the Mx, ORS and DP areas in the bulb and the basal and suprabasal layers in the bulge using ImageJ software.”

5. In Figure 3, authors used TYR and TRP1 staining to detect melanocytes. These two enzymes are markers for differentiated melanocytes. Other markers (such as Mitf, TRP2, etc.) might help to demonstrate whether undifferentiated melanocytes are still present there. In addition, did the melanocytes undergo apoptosis?

Response: According to your input, we tried to show the response of undifferentiated melanocytes with costaining for MITF and cleaved caspase-3 in the bulge of HFs. However, it

was very difficult to detect MITF⁺ cells in the bulge of in vivo HF xenografts (Figure S1). Considering that the supply dynamics from undifferentiated cells to fully differentiated cells of the melanocyte lineage are very similar to those of the epithelial lineage in HFs, we suggest that follicular melanocytes and undifferentiated melanocytes follow the apoptotic dynamics similar to those of hair matrix cells and hair follicle stem cells, respectively.

6. In Figure 4d, in vitro testing showed that Bu treatment showed dystrophic changes of hair bulb, but Bu/Cy treatment did not lead to hair bulb dystrophy. Could the authors examine the histology and cell proliferation/apoptosis of the hair bulbs? This can help to verify whether the in vitro model is consistent with the humanized mouse model.

Response: We regret that the previous description of Figure 4d (in the revised manuscript) was not appropriate to describe the exact phenotype of Bu/Cy-treated HFs. In ex vivo HF organ cultures, Bu-treated HFs prematurely entered the dystrophic catagen pathway without p53-dependent apoptosis in their epithelial strands, suggesting that HF epithelial cells are still biologically viable enough to progress into their response pathway. However, Bu/Cy-treated HFs showed a completely shrunken bulb morphology up to the bottom end of hair shaft, indicating a complete loss of the bulb and cellular arrest of the HF epithelium. We also added Figure S3 to show the spatiotemporal response with immunostaining of Ki67⁺K15⁺ cells and cleaved caspase-3⁺K15⁺ cells in the bulb area of organ-cultured HFs after Bu/Cy treatment.

7. In Figure S1, "Bu-induced proliferation in the basal layer in the bulb area". In this figure, the indicated proliferating cells (white arrows) are mostly in the ORS above the hair bulb. It is not consistent with the title of this figure.

Response: We are grateful for this detailed comment. The title of Figure S2 was revised to "Bu-induced proliferation of basal ORS cells". Thank you.

8. In Figure S4. The cells do not seem to be confluent in culture. IF authors meant to culture holoclones-rich ORS cells into confluency and tested the effect of Bu, Cy, Bu/Cy, how was 100% confluency determined?

Response: Human ORS cells derived from the bulge areas were allowed to reach 100% confluence and cultured for an additional 2 days. We added the microscopic figures to show

the 100% confluent population and its changes after Bu and/or Cy treatment. The floating cells were distinguishable by their free movement in culture medium and outlined with a red-dotted line. However, cells did not grow completely evenly in every part of the dish, so we were able to find a few spots with less than 100% confluency. Morphological changes of the cells are more obvious in these spots because we can see their cellular membrane structure easily. Cell confluency was assessed by microscopic observation, and the cell cycle was confirmed by flow cytometry analysis.

9. There are a couple of grammatical errors in the manuscript. Authors are suggested to go over their manuscript carefully to correct these errors.

Response: The English in the revised version of the manuscript has been checked by a professional English editing service provided by Springer Nature Author Services.

Reviewer #2 (Remarks to the Author):

The topic studied here is of great clinical relevance as there is an increasing number of cases where alopecia after chemotherapy (CIA) is permanent, suggesting major, irreversible stem cell damage. Yet, the latter has been poorly investigated. Therefore, the authors are to be commended for tackling this major unsolved problem in clinical oncology by using the instructive & clinically relevant humanized mouse CIA model they had developed & published before (JID 2016).

That damaged human bulge stem cells initially undergo (under inflammatory, IFN γ -driven conditions) proliferation and subsequently are driven into apoptosis within their stem cell niche has already been documented for another form of human permanent alopecia, lichen planopilaris (Harries et al. J Pathol 2013). This has invited the "stem cell exhaust" hypothesis in the pathobiology of permanent alopecia (Harries et al. Trends Mol Med 2018). The same phenomenon is seen in the rapidly proliferating progeny of human bulge stem cells, i.e. in the transit amplifying cells of the anagen hair bulb under conditions of chemotherapy in organ-cultured human scalp HFs (Bodo et al. AJP 2007). Surprisingly, neither of these previously reported observations, which are directly relevant in the current context, are properly cited and discussed here.

Response: We regret that we missed the previous important studies and appreciate the

constructive comment for this study. According to the inputs from reviewers 1 and 2, we added a discussion about the possibility of a generalized mechanism for permanent HFSC loss as follows: "Stepwise mechanisms consisting of initial proliferation and subsequent apoptosis can be found in other contexts in HF biology. The activation of HFSCs was also observed in the dystrophic anagen response after radiotherapy, and the final response to ionizing radiation was suggested to be decided according to the activation state of HFSCs. In lichen planopilaris, which is interferon-gamma (IFN γ)-driven inflammatory permanent hair loss, HFSCs initially underwent proliferation and subsequently underwent apoptosis within their stem cell niche. Considering the frequent observation of the cellular dynamics relevant in this study, a priming stem cell activation and timely induction of a successive insult is suggested to be a general mechanism that effectively leads to stem cell exhaustion not only after exposure to genotoxic agents also under inflammatory conditions."

That chemotherapy can induce permanent alopecia with the morphological correlate of HF deletion (incl. loss of the so-called permanent part of the HF) is well-known and has been described in many clinical and limited dermatopathological reports and is thus not novel either. The same applies to the finding that the cellular response to alkylating agents is highly cell cycle status-dependent. In any case, the data provided here regarding the proliferation-dependence of human HF stem cell responses to alkylating agents are only correlative and attempts to selectively arrest the HF stem cells in defined cell cycle phases so as to observe how this differentially impacts on their chemotherapy response were not made.

Conceptually, this appears to restrict the novelty of the findings reported here to the observation that HFSC proliferation presumably was activated through the PI3K/Akt pathway, and that depletion may have been driven by p38-dependent cell death. This is interesting since CIA-associated apoptosis, at least in the hair matrix, is widely believed to be p53-dependent (Botchkarev et al. *AJP* 2000). However, this dogma has previously been challenged by findings in chemotherapy-treated feather follicles, where the dominant molecular matrix keratinocyte response is the down-regulation of Shh, not p53-dependent apoptosis (Xie et al. *JID* 2015) - another study that the authors may wish to consult and discuss in the context of their findings. Regrettably, the role of Shh in the events leading up to human bulge epithelial stem cell apoptosis was not investigated.

In any case, the postulated role of PI3K/Akt in stem cell proliferation, and of p38 in stem cell apoptosis is only assumed on the basis of correlative data, but not definitively proven. Global

gene expression profiling of the human bulge in response to chemotherapy during different time points, using laser capture microdissection, which has been used before when investigating human permanent alopecia (Harries et al. 2013, Imanishi et al. JID 2018), would be an excellent method for interrogating the molecular damage response pathways of bulge stem cells in situ much more instructively and comprehensively. Also, a number of techniques have been published that permit one to isolate or at least enrich for human bulge-derived epithelial stem cells. Using one of these human HF stem cell culture methods, mechanistic studies would seem possible through which the authors' PI3K/Akt and p38 hypotheses could likely be verified.

Response: We agree with the reviewer's comments and regret that the original description was not sufficient to clarify how human HFSCs undergo proliferation and apoptosis after Bu/Cy treatment. According to the reviewer's comments, we performed additional experiments to determine the molecular mechanism of permanent loss of HFSCs and the global gene expression changes in HFSCs using laser capture microdissection after Bu/Cy treatment.

1. Role of PI3K/Akt pathway activation and the p53/p38-induced apoptosis cascade

To determine the temporal dynamics after sequential Bu/Cy treatment, ORS cells were consecutively harvested after 0, 1, 3, and 6 h of priming Bu treatment and then after 1, 3, 6, and 12 h of subsequent Cy treatment (Figure 6b). Protein analysis showed dramatic changes in two different phases: the PI3K/Akt pathway activation with upregulated cyclin D1 after Bu treatment, and the subsequent conversion into p53/p38-induced cell death with PI3K/Akt pathway inhibition after Bu/Cy treatment, resulting in dynamic changes in p21, representing the cellular decision of cell survival or cell death (Figure 6d).

To confirm the phase conversion of the DNA damage response, we performed immunostaining for the phosphorylated Akt and p38 proteins in the *in vivo* HF xenografts. Consistent with the protein analysis, Akt phosphorylation occurred after Bu treatment (Figure 7a), and p38 phosphorylation occurred after Bu/Cy treatment (Figure 7b) in the K15+ HFSCs.

Then, we investigated the roles of the PI3K/Akt pathway and the p38-induced cascade with specific inhibitors of PI3K (20 μ M; LY294002) and p38 (10-20 μ M; SB202190 or SB203580). Under treatment with the PI3K inhibitor, Akt phosphorylation was completely blocked, with loss of cyclin D1 induction (Figure 7c). Notably, p53 and phosphorylated p38 were upregulated during blockade of the PI3K/Akt pathway, supporting the hypothesis that

the PI3K/Akt pathway is the active part of the DNA repair process. Unexpectedly, the driving force of p38 phosphorylation after alkylating chemotherapy was too strong to be blocked by treatment with p38 inhibitors (Figure 7d). Overwhelming potentiation of p38 phosphorylation was observed, with a DNA damage severity-dependent downregulation of p21 and a marginal increase in cleaved caspase-3.

The roles of the PI3K/Akt pathway and p53/p38-induced cascade were clearly observed during phase conversion after sequential Bu/Cy treatment (Figure 7e). The temporal changes in p53 combined with other proteins directly demonstrated two points: the first is that DNA damage sensitivity remains low if there is no priming proliferation by a PI3K inhibitor, and the other is that cells with DNA damage are not efficiently eliminated by insufficient p38 activation even at the last timepoint.

2. Shh signaling pathway involvement or changes

We appreciate the reviewer for making this insightful comment. We found a previous article about the role of Shh signaling in chemotherapy-treated feather follicles J Invest Dermatol 135, 690-700 (2015). However, there is limited information regarding how Shh signaling is regulated or its role in human HFSCs. In the quiescent state of human HFSCs, Sonic hedgehog (Shh) signaling blocks differentiation into progeny in quiescent HFSCs in humans Aging Cell 8, 738-751 (2009). Consistent with previous reports, Shh expression was observed in most HFSCs in the basal layer in normal HFs and another cell population, the second progeny in the suprabasal layer, which is more differentiated than the first progeny of HFSCs J Clin Invest 104, 855-864 (1999). Interestingly, Shh⁺ cells almost disappeared after Bu treatment and then reappeared without K15 expression in the basal layer after Bu/Cy treatment (Figure 3f). Together with the results of Lhx2 expression, these findings suggest that HFSCs losing their quiescence after Bu treatment show a differentiated phenotype in addition to massive apoptosis after Bu/Cy treatment, indicating loss of the stem cell reserve.

3. Global gene expression profiling using laser capture microdissection of HFSCs

To consolidate the HFSC response in permanent CIA in humans, global gene expression changes were evaluated in the in vivo HF xenografts after sequential Bu/Cy treatment. Using laser capture microdissection, HFSCs were obtained from the basal bulge

layer and analyzed with RNA-seq (Figure S6). Based on hierarchical clustering (Figure 8a), the upregulated genes after Bu or Bu/Cy treatment were selected for functional enrichment pathway analysis from the Reactome database. Mitotic cell cycle pathways and responsible genes were upregulated after Bu, accompanied by the DNA repair pathway and p53 transcriptional regulation (Figure 8b and S7a). Intriguingly, cell cycle checkpoint pathways, especially G2/M checkpoints and responsible genes, were upregulated after Bu/Cy treatment, along with mitochondrial membrane-associated pathways (Figure 8c and S7b). This result indicates that the majority of HFSCs experienced mitotic catastrophe and stalled in S phase with activation of the G2/M checkpoint, which is consistent with the S phase arrest observed during the ORS cell response according to proliferation status ^{Nat Rev Cancer 7, 861-869 (2007). Oncogene 23, 2825-2837 (2004)}. Gene expression of Ki67 (MKI67) and Survivin (BIRC5), an apoptosis inhibitor, was upregulated after Bu treatment and then decreased after Bu/Cy treatment (Figure 8d). Genes for maintaining HFSC stemness (LHX2, PHLDA1, FZD1, and TGFB2) were already downregulated after Bu treatment, supporting the role of priming of proliferation in loss of the stem cell reserve.

p53-dependent apoptosis is widely believed to be an essential player for the preservation of genomic integrity in response to DNA damage in the HF epithelium. In addition, p38 kinase activation is induced by DNA cross-linking agents and is sustained for more than a few days, triggering the apoptotic cascade ^{Mol Biol Cell 14, 2071-2087 (2003). Trends Mol Med 12, 440-450 (2006)}. p38-induced apoptosis can be driven not only in a p53-dependent manner ^{EMBO J 18, 6845-6854 (1999)} but also by a p53-compromised state that originates from the overwhelming extent of DNA damages ^{Mutat Res 836, 89-97 (2018)}. More interestingly, initiation of the G2/M checkpoint requires the activation of p38 kinase, and this p38-mediated G2/M checkpoint also occurs in p53-deficient cells ^{Nature 411, 102-107 (2001)}. In this study, it is thought that p53 activation is partially responsible for the apoptotic loss of HFSCs during the early period after DNA damage, but sustained p38 activation augmented the resulting apoptosis as a major mediator of mitotic catastrophe after irreversible DNA damage.

Finally, have the authors considered how one - potentially important - methodological weakness of their CIA model, namely the transplantation of isolated scalp HFs rather than of full-thickness human scalp skin - might have impacted on the results they obtained? There is increasing appreciation that human HFs operate in the context of a complex functional skin

appendage unit (i.e. HF+ arrector pili muscle + sebaceous gland + eccrine gland coil + a cone of dermal adipocytes that enwraps all of these structures, incl. the bulge [Poblet et al. BJD 2018]). Since all of these human tissues/cells are missing in their hair xenotransplant model (Yoon et al. JID 2016), one wonders to which extent the absence of this distinct human peribulge tissue signaling milieu may have exaggerated, disrupted or distorted the HF stem cell responses to chemotherapy observed here. It would seem appropriate to at least discuss this possibility.

Response: We appreciate this constructive comment. We added the following discussion: "Even though the permanent CIA model in this study clarified the spatiotemporal dynamics of HFSCs after chemotherapy, the limitation is that the macroenvironment of isolated human HFs was supported by mouse skin, not by human scalp skin. Considering that human HFs operate within the complex functional skin appendage unit, the complete elimination of HF xenografts in this model is thought to be exaggerated compared to the partial elimination of HFs in patients due to differential interfollicular susceptibility to chemotherapy."

Reviewer #3 (Remarks to the Author):

This is a very interesting article and deserves publishing.

1. I would like to see in discussion speculations on other classes of drugs that may follow similar mechanism of effect in hair follicle.

Response: We appreciate this insightful input. According to the reviewer's comments, we added a discussion to suggest the stepwise process of initial proliferation and subsequent apoptosis as a generalized mechanism for permanent HFSC loss as follows: "Stepwise mechanisms consisting of initial proliferation and subsequent apoptosis can be found in other contexts in HF biology. The activation of HFSCs was also observed in the dystrophic anagen response after radiotherapy, and the final response to ionizing radiation was suggested to be decided according to the activation state of HFSCs. In lichen planopilaris, which is interferon-gamma (IFN γ)-driven inflammatory permanent hair loss, HFSCs initially underwent proliferation and subsequently underwent apoptosis within their stem cell niche. Considering the frequent observation of the cellular dynamics relevant in this study, a priming stem cell activation and timely induction of a successive insult is suggested to be a general mechanism that effectively leads to stem cell exhaustion not only after exposure to genotoxic agents also under inflammatory conditions."

2. Please expand and comment on the importance of timing of successive insult by the chemotherapeutic agents for permanency of alopecia - in humans.

Response: We appreciate this constructive comment. Based on cell cycle analysis and protein analysis with global gene expression changes, we found that the majority of HFSCs experienced mitotic catastrophe and added a discussion to emphasize that point, as follows: "Considering the frequent observation of the cellular dynamics relevant in this study, a priming stem cell activation and timely induction of a successive insult is suggested to be a general mechanism that effectively leads to stem cell exhaustion not only after exposure to genotoxic agents also under inflammatory conditions. Moreover, it is supposed that permanent hair loss caused by other chemotherapeutic regimens that require time-dependent combination may be driven by a similar mechanism, including mitotic catastrophe." Due to the large number of additional experiments and new findings, it was very hard to add a more detailed discussion of this suggestion. Thank you so much for your input.

REVIEWERS' COMMENTS:

Reviewer #1 (Remarks to the Author):

For this revision, authors performed many new experiments. They included many new data and revised the manuscript extensively. The quality of this manuscript has been improved significantly. Their new data support their claims/conclusion. My concerns were all addressed in the rebuttal letter and in the revised manuscript.

I have one minor suggestion:

1. In revised Figure 9, I'll suggest authors point out dermal papilla cells in the depiction (brown cells in the depiction). Because the data from the manuscript did not strongly support that chemotherapy induces dermal papilla cell death, I suggest authors put a question mark here or leave whether dermal papilla cells die open.

Sung-Jan Lin

Reviewer #2 (Remarks to the Author):

I commend the authors for the thorough and overall convincing revision of their study, and have only a few minor suggestions for manuscript optimization.

1. Careful language editing is recommended.

2. Figs. 1, 6, 7 contain largely or exclusively qualitative data, which makes it very difficult to judge how robust, representative, and reproducible the reported results really were. Since this is obviously important, the authors are strongly encouraged to show, wherever this is possible possible but not yet the case, corresponding quantitative data, at least in the supplement.

3. Where this information is still missing, the authors should carefully indicate in each figure legend the n of independent experiments/mice that underlies the reported data sets (currently, this often remains somewhat unclear).

4. The cartoon (Fig. 9) and the corresponding text segment in the Discussion question whether the authors have fully understood the concept of the "dystrophic anagen" versus "dystrophic catagen" pathway response of hair follicles to chemotherapy-induced damage (the two distinct pathways are mislabeled in the cartoon, and the legend fails to indicate on which prior publication the basic design of this cartoon is based [possibly, Paus et al. Lancet Oncol 2013 ?]; both needs to be corrected/improved accordingly).

Also, this cartoon would be a lot more instructive, if it also clearly indicated selected key molecular drivers in the sketched hair follicle damage pathways that the authors have identified here, and if the choreography of proposed molecular pathobiology events is at least hinted at.

5. In the revised Discussion, I'd enjoy reading at least a short comment on the recent Huang WY et al. Exp Dermatol. 2019 paper in the light of the authors' findings.

Congratulations on a great study!

We are pleased to submit a revised version of our manuscript entitled "Priming mobilization of hair follicle stem cells triggers permanent loss of regeneration after alkylating chemotherapy" for consideration for the publication in Nature Communications. We appreciate the referees for their constructive inputs and positive feedback. We have carefully examined the reviewers' comments and addressed the reviewers' suggestions by providing point-by-point responses.

REVIEWERS' COMMENTS:

Reviewer #1 (Remarks to the Author):

For this revision, authors performed many new experiments. They included many new data and revised the manuscript extensively. The quality of this manuscript has been improved significantly. Their new data support their claims/conclusion. My concerns were all addressed in the rebuttal letter and in the revised manuscript.

Response: We are glad to have a constructive revision thanks to the reviewer's insightful comments.

I have one minor suggestion:

1. In revised Figure 9, I'll suggest authors point out dermal papilla cells in the depiction (brown cells in the depiction). Because the data from the manuscript did not strongly support that chemotherapy induces dermal papilla cell death, I suggest authors put a question mark here or leave whether dermal papilla cells die open.

Response: We agree with the reviewer and erased any meaningful marks on dermal papilla cells in Fig. 9 to leave whether dermal papilla cells die open.

Sung-Jan Lin

Reviewer #2 (Remarks to the Author):

I commend the authors for the thorough and overall convincing revision of their study, and have only a few minor suggestions for manuscript optimization.

Response: We are glad to have a constructive revision thanks to the reviewer's insightful comments.

1. Careful language editing is recommended.

Response: The English in the previous version of the manuscript was checked by a professional English editing service provided by Springer Nature Author Services, and the current revised version of the manuscript has been checked again by the native American scientist.

2. Figs. 1, 6, 7 contain largely or exclusively qualitative data, which makes it very difficult to judge how robust, representative, and reproducible the reported results really were. Since this is obviously important, the authors are strongly encouraged to show, wherever this is possible but not yet the case, corresponding quantitative data, at least in the supplement.

Response: We agree with the reviewer and added the quantification plots as Supplementary Fig. 7 for Fig. 6b, Supplementary Fig. 8 for Fig. 7c and 7d, and Supplementary Fig. 9 for Fig. 7e including Fig. 6d. For Fig. 1 which only contains largely qualitative data, it is impossible to include the quantitative information.

3. Where this information is still missing, the authors should carefully indicate in each figure legend the n of independent experiments/mice that underlies the reported data sets (currently, this often remains somewhat unclear).

Response: We regret that the n description was not clear enough to provide the exact information of performed experiments. The description for the n was changed to provide the precise number of biological replicates/timepoint, biological replicates/group, and independent experiments in each figure legend according to the editor's suggestion. We erased the n descriptions presented as a range and changed the n number for Fig. 4b and 4c and define the n number for Fig. 8d.

4. The cartoon (Fig. 9) and the corresponding text segment in the Discussion question whether the authors have fully understood the concept of the "dystrophic anagen" versus "dystrophic catagen" pathway response of hair follicles to chemotherapy-induced damage (the two distinct pathways are mislabeled in the cartoon, and the legend fails to indicate on which prior publication the basic design of this cartoon is based [possibly, Paus et al. Lancet Oncol 2013 ?]; both needs to be corrected/improved accordingly).

Also, this cartoon would be a lot more instructive, if it also clearly indicated selected key molecular drivers in the sketched hair follicle damage pathways that the authors have identified here, and if the choreography of proposed molecular pathobiology events is at least hinted at.

Response: We are grateful for this constructive comment. We changed the labeling of dystrophic anagen or catagen pathway in their original way and added the prior publication regarding the basic concepts of those pathway and the application to human HF biology. We also added the key molecular pathway identified in this study into Fig. 9.

5. In the revised Discussion, I'd enjoy reading at least a short comment on the recent Huang WY et al. Exp Dermatol. 2019 paper in the light of the authors' findings.

Response: We added the discussion that the regenerative attempt of bulge stem cells shows slow activation dynamics after severe DNA damage as the below:

“The reactive proliferation of HFSCs can be understood as cellular attempts at DNA repair and tissue regeneration. The PI3K/Akt pathway, responsible for the mobilization of HFSCs⁵⁷, is a direct participant coregulated tightly with other components of DNA damage repair pathways^{36,58}. For tissue regenerative attempts, mouse adult HFSCs exhibit slower activation dynamics only when HFs are severely damaged after ionizing radiation⁵⁹.”

Congratulations on a great study!